# MOMENT MATTERS: MEAN AND VARIANCE CAUSAL GRAPH DISCOVERY FROM HETEROSCEDASTIC DATA

## ABSTRACT

This paper proposes a Bayesian causal discovery approach to uncover the causal mechanisms underlying heteroscedasticity, where the variance of one variable is influenced by the values of the others. To distinguish between the causes that affect the mean and those that influence the variance, we infer the posterior distribution over *mean* and *variance causal graphs*, whose structures can be different, depending on the moment information. We establish identifiability conditions for these causal graphs by extending the results on heteroscedastic noise models (HNMs). Building on these conditions, we develop a variational inference framework that can incorporate prior knowledge about the node orderings of the underlying graphs. We experimentally show that our method can successfully infer both mean and variance causal graphs, outperforming the state-of-the-art baselines.

## 1 INTRODUCTION

Many scientists are actively striving to uncover the causal mechanisms that govern complex real-world phenomena. Causal discovery contributes to this goal by inferring cause-effect relationships between variables as a *causal graph*, whose edge represents that one variable changes another variable values. However, such edges do not specify which statistical moments are affected by each cause, thereby limiting interpretability in complex systems that exhibit *heteroscedasticity*, where different causes may influence the (conditional) mean and variance of a variable.

**Motivating Example:** Consider a drug engineer who refines a compound to achieve more consistent therapeutic effects by reducing variability in a target protein's activity across individuals (Michoel and Zhang, 2023). By inferring the subgraph of underlying causal structure shown in Figure 1 (a), the engineer selects protein $X_1$ as the downstream target and plans to intervene on its causes, $X_2, \ldots, X_6$, which are pharmacologically manipulatable regulators. However,

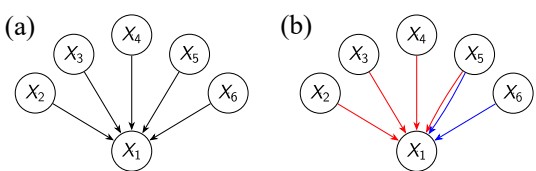

Figure 1: (a) Moment-agnostic causal graph and (b) mean and variance causal graphs (red and blue).

this standard causal graph is unsatisfactory: Although it narrows down the candidate causes of $X_1$, it does not reveal which causes influence $X_1$'s variance, since such a *moment-agnostic* causal graph does not distinguish the causes that affect the mean and those that influence the variance. By contrast, if the engineer could obtain *mean* and *variance causal graphs*, which make such distinctions as shown in Figure 1 (b), they could design experiments by first focusing on achieving the desired mean of $X_1$ and subsequently varying the interventions on $X_5$ and $X_6$ to reduce $X_1$'s variance while preserving its mean. This targeted strategy would greatly facilitate the drug design process.

Beyond this example, the three real-world domains highlight the importance of identifying and controlling the drivers of heteroscedasticity, i.e., causes that affect the (conditional) variance of variables. First, in *systems biology*, whereas most molecules mainly shift mean expression, certain factors (e.g., stress-response proteins) modulate variance (Guilbert et al., 2020; Boopathy et al., 2022); detecting such modulators deepens our understanding of the regulatory basis of cell-to-cell variability (Daye et al., 2012; Oyarzun et al., 2015). Second, in *economics*, it is of great interest to stabilize key economic outcomes by controlling their cross-sectional variance (Braumoeller, 2006). Third, in *algorithmic fairness*, the goal is to achieve equitable outcomes (e.g., in hiring, lending) with

respect to sensitive attributes (e.g., gender, race, age, disability status) (Kusner et al., 2017; Wu et al., 2019; Chikahara et al., 2021; 2023; Zuo et al., 2024). To promote forward-looking, fair decision-making, it is crucial to identify *latent sensitive attributes*, i.e., features not legally designated as sensitive yet linked to disparities and controversial to use. Detecting variance-level causes is helpful for this purpose, as decision rules that avoid subgroups with high outcome variance for risk aversion may induce particularly latent forms of statistical discrimination (Dickinson and Oaxaca, 2009).

These practical real-world scenarios motivate a key research question: *Can we separately identify the mean and variance causal graphs solely from observational data, collected without intervention?* Our answer is yes. By extending the results on *heteroscedastic noise models* (HNMs) (Khemakhem et al., 2021; Yin et al., 2024), we theoretically derive the identifiability conditions for mean and variance causal graphs, whose structures may differ, depending on the moment information.

Once we have established theoretical identifiability, we turn to the practical challenge of inferring these two separate graphs from finite data. Yet point estimation of each graph can be unreliable, especially under small-sample scenarios in scientific fields like medicine (Wiens et al., 2014). This practical requirement raises a second research question: *How can we quantify the inference uncertainty of mean and variance causal graphs under data scarcity?* To answer this, building on the recent variational inference technique (Charpentier et al., 2022), we establish a *Bayesian causal discovery* approach that jointly infers the posterior distribution over the mean and variance causal graphs. This approach enables us to quantify the inference uncertainty by computing the posterior probabilities for arbitrary structural features, such as the presence of an edge, a path, or a subgraph, thereby facilitating the discovery of highly probable cause-effect relationships from finite data (Friedman and Koller, 2003). We demonstrate the practical utility of this Bayesian approach by conducting a real-world case study in Section 5.2, where our method successfully identifies the biologically plausible variance-controlling relationship between signaling proteins from limited data.

**Our contributions** are summarized as follows:

- We propose a *mean-variance HNM*, which is associated with mean and variance causal graphs (Section 3.1), and theoretically derive their identifiability conditions (Section 3.2).
- We establish a variational inference framework that learns the posterior over the mean and variance causal graphs (Sections 4.1 and 4.2) and develop a prior knowledge incorporation approach that can utilize the domain knowledge about the node orderings in causal graphs (Section 4.3).
- We experimentally show that our proposed framework can successfully discover the mean and variance causal graph structures (Section 5.1), and can accurately infer moment-agnostic causal graphs under small-sample scenarios, by leveraging prior knowledge (Section 5.2).

## 2 PRELIMINARIES

**Structural Causal Model and Causal Graph:** A structural causal model (SCM) (Pearl, 2009) describes a data-generation process over a set of *endogeneous variables* $\boldsymbol{X}$, whose causal relationships are of interest. Formally, given a set of other random variables called *exogenous variables* $\boldsymbol{E}$ (e.g., noise variables) and a set of deterministic functions $\mathcal{F}$, an SCM defines a *structural equation*, which determines the values of each $X_j \in \boldsymbol{X}$ ($j = 1, \ldots, d$) as the output of deterministic function $f_j \in \mathcal{F}$:

$$X_j = f_j(\boldsymbol{X}_{\mathrm{pa}(j)}, \boldsymbol{E}_j) \quad \text{for} \quad j = 1, \ldots, d, \tag{1}$$

where $\boldsymbol{X}_{\mathrm{pa}(j)} \subseteq \boldsymbol{X} \backslash X_j$ and $\boldsymbol{E}_j \subseteq \boldsymbol{E}$ are subsets of endogenous and exogenous variables. Variables $\boldsymbol{X}_{\mathrm{pa}(j)}$ directly affect $X_j$'s values and are hence referred to as $X_j$'s *parents*.

Causal graph $G$ encodes these parental relationships: It has edge $X_i \to X_j$ for $i, j \in \{1, \ldots, d\}$ and $i \neq j$ if and only if $X_i \in \boldsymbol{X}_{\mathrm{pa}(j)}$. We define *adjacency matrix* $\mathbf{A} \in \{0, 1\}^{d \times d}$ of $G$ by $A_{i,j} = 1$ if $X_i \to X_j$; otherwise, $A_{i,j} = 0$. In this paper, we call $G$ a moment-agnostic causal graph, as it does not distinguish causes by the statistical moment (e.g., mean or variance) they influence.

In general, from observational data alone, a causal graph can be inferred only up to the Markov equivalence class (MEC), i.e., the class of causal graphs that encode identical conditional independence relations to joint distribution $\mathrm{P}(\boldsymbol{X})$. However, if the data are generated by an SCM belonging to a certain restricted functional class, the causal graph can be uniquely identified. Among such identifiable SCM classes, we focus on the one with weak assumptions, called HNMs.

**Heteroscedastic Noise Model (HNM):** An SCM follows an HNM if the structural equation obeys

$$X_j = m_j(\boldsymbol{X}_{\mathrm{pa}(j)}) + v_j(\boldsymbol{X}_{\mathrm{pa}(j)})E_j \quad \text{for } j = 1, \ldots, d, \tag{2}$$

where $m_j, v_j \in \mathcal{F}$ are deterministic functions with $v_j(\cdot) > 0$ for any input, and $E_j \in \boldsymbol{E}$ is the noise that has zero mean ($\mathbb{E}[E_j] = 0$) and is mutually independent ($E_i \perp\!\!\!\perp E_j$ for $i \neq j$) (Xu et al., 2022).

A notable advantage of HNMs is that conditional variance $\mathbb{V}[X_j \mid \boldsymbol{X}_{\mathrm{pa}(j)}] = \mathbb{V}[E_j] \left(v_j(\boldsymbol{X}_{\mathrm{pa}(j)})\right)^2$ is given as a function of $\boldsymbol{X}_{\mathrm{pa}(j)}$, thus capturing heteroscedasticity in complex real-world data (e.g., gene expression data (Imoto et al., 2003)). This ability to represent complex heteroscedastic relationships sharply contrasts with two identifiable SCM classes. Additive noise models (ANMs) (Shimizu et al., 2006; Hoyer et al., 2008) assumes homoscedasticity, as their structural equation is given as a special case of HNMs in (2) with constant $v_j(\cdot)$. Although post-nonlinear models (PNLs) (Zhang and Hyvärinen, 2009) are more general than ANMs and can capture multipricative noise unlike HNMs, they also assume homoscedasticity and thus cannot detect the source of variability (Yin et al., 2024).

For this reason, there is a growing interest in HNMs. Early work have shown bivariate identifiability (Khemakhem et al., 2021; Xu et al., 2022), which is extended to multivariate setup (Strobl and Lasko, 2023; Yin et al., 2024). Building on their identifiability results, likelihood-based methods (Immer et al., 2023; Kikuchi, 2022; Tran et al., 2024; Yin et al., 2024) and independence-test-based methods (Duong and Nguyen, 2023; Immer et al., 2023; Lin et al., 2025) have been proposed for the point estimation of the causal graph. Our work differs in two key aspects. (i) Our inference targets are mean and variance causal graphs, whose definition and identifiability are presented in Section 3. (ii) Unlike existing point estimation methods, we develop a Bayesian approach that infers the posterior distribution over causal graphs (Section 4). This enables Monte Carlo approximation of posterior probabilities for structural features, such as the presence of an edge, a path, or a subgraph (Giudici and Castelo, 2003), thus facilitating the discovery of highly probable structural features.

## 3 Towards Moment-Driven Causal Graph Discovery

As a novel class of SCMs, we propose *mean-variance HNMs* with mean and variance causal graphs in Section 3.1. We then discuss the identifiability conditions of these causal graphs in Section 3.2.

### 3.1 Mean-Variance HNM with Mean and Variance Causal Graphs

To accommodate possibly distinct mean- and variance-level causal relationships, we define a mean-variance HNM of each variable $X_j \in \boldsymbol{X}$ by modifying the original HNM in (2) as

$$X_j = m_j\left(\boldsymbol{X}_{\mathrm{pa}^M(j)}\right) + v_j\left(\boldsymbol{X}_{\mathrm{pa}^V(j)}\right) E_j \quad \text{for } j = 1, \ldots, d, \tag{3}$$

where $\boldsymbol{X}_{\mathrm{pa}^M(j)}, \boldsymbol{X}_{\mathrm{pa}^V(j)} \subseteq \boldsymbol{X} \backslash X_j$ are variable subsets, $m_j, v_j$ are functions with $v_j(\cdot) > 0$, and $E_j$ is a noise with $\mathbb{E}[E_j] = 0$ and $E_i \perp\!\!\!\perp E_j$ for $i \neq j$. As with Duong and Nguyen (2023); Yin et al. (2024), we assume that $E_j$ follows a (zero-mean) Gaussian distribution. Under this assumption, statistical moments affected by $\boldsymbol{X}_{\mathrm{pa}(j)} \coloneqq \boldsymbol{X}_{\mathrm{pa}^M(j)} \cup \boldsymbol{X}_{\mathrm{pa}^V(j)}$ are restricted to mean and variance, which are determined using functions $m_j$ and $v_j$ as $\mathbb{E}[X_j \mid \boldsymbol{X}_{\mathrm{pa}(j)}] = m_j(\boldsymbol{X}_{\mathrm{pa}^M(j)})$ and $\mathbb{V}[X_j \mid \boldsymbol{X}_{\mathrm{pa}(j)}] = \mathbb{V}[E_j] \left(v_j(\boldsymbol{X}_{\mathrm{pa}^V(j)})\right)^2$. We hence refer to $m_j$ and $v_j$ as *mean* and *variance functions*.

We define the mean and variance causal graphs based on parental variables $\boldsymbol{X}_{\mathrm{pa}^M(j)}$ and $\boldsymbol{X}_{\mathrm{pa}^V(j)}$: Mean causal graph $G^M$ has edge $X_i \to X_j$ if and only if $X_i \in \boldsymbol{X}_{\mathrm{pa}^M(j)}$, and variance causal graph $G^V$ contains edge $X_i \to X_j$ if and only if $X_i \in \boldsymbol{X}_{\mathrm{pa}^V(j)}$. We define their adjacency matrices $\mathbf{A}^M, \mathbf{A}^V \in \{0, 1\}^{d \times d}$ by $A_{i,j}^M = 1$ if $X_i \to X_j$ in $G^M$; otherwise, $A_{i,j}^M = 0$; $\mathbf{A}^V$ is defined similarly. We also introduce moment-agnostic causal graph $G$, whose adjacency matrix $\mathbf{A} \in \{0, 1\}^{d \times d}$ is given by taking the logical OR of binary matrices $\mathbf{A}^M$ and $\mathbf{A}^V$ as $A_{i,j} = A_{i,j}^M \vee A_{i,j}^V$.

**Connection to Original HNM:** When the input sets are identical ($\boldsymbol{X}_{\mathrm{pa}^M(j)} = \boldsymbol{X}_{\mathrm{pa}^V(j)} = \boldsymbol{X}_{\mathrm{pa}(j)}$), our mean-variance HNM reduces to the original HNM. Conversely, any instance of a mean-variance HNM can be rewritten as the original HNM via input masking: The mean and variance functions in the former can be reformulated as $m_j(\boldsymbol{S}_{\mathrm{pa}^M(j)}(\boldsymbol{X}_{\mathrm{pa}(j)}))$ and $v_j(\boldsymbol{S}_{\mathrm{pa}^V(j)}(\boldsymbol{X}_{\mathrm{pa}(j)}))$, where $\boldsymbol{S}_{\mathcal{I}} \colon \boldsymbol{X}_{\mathrm{pa}(j)} \to \boldsymbol{X}_{\mathcal{I}}$ is a masking function that selects the variables indexed by $\mathcal{I} \subset \{1, \ldots, d\}$ from parents $\boldsymbol{X}_{\mathrm{pa}(j)}$ in moment-agnostic graph $G$. Thus, two HNMs expresses the same class of functions.

Such functional equivalence implies that, to identify mean and variance causal graphs $G^M$ and $G^V$, we must identify not only the moment-agnostic causal graph $G$ of the original HNM but also the variable selection mechanism for $G^M$ and $G^V$. For this reason, we build on the existing results for the original HNM (Khemakhem et al., 2021; Yin et al., 2024) to derive the identifiability conditions.

### 3.2 IDENTIFIABILITY RESULTS

Our identifiability results rely on four technical assumptions (see Appendix A for details).

**Assumption 3.1** (Causal sufficiency)**:** *Exogenous noises satisfy $E_i \perp\!\!\!\perp E_j$ for any $i, j \in \{1, \ldots, d\}$.*

**Assumption 3.2** (Causal minimality)**:** *Joint distribution $\mathrm{P}(\boldsymbol{X})$ satisfies the causal minimality condition with respect to moment-agnostic causal graph $G$.*

**Assumption 3.3:** *Mean and variance graphs $G^M$ and $G^V$ are directed acyclic graphs (DAGs).*

**Assumption 3.4** (Shared permutation condition)**:** *There exists an identical permutation (a.k.a., topological ordering) of mean and variance causal graphs $G^M$ and $G^V$.*

In Assumption 3.4, permutation $\pi : \{1, \ldots, d\} \to \{1, \ldots, d\}$ of a DAG is a node ordering such that no node has a directed path to any node that precedes it in the order; that is, if $\pi(i) < \pi(j)$, then $X_{\pi(j)}$ cannot have a directed path to $X_{\pi(i)}$. This ordering is **not** unique because a DAG admits multiple permutations. Sharing a permutation between two DAGs implies that their *union* (i.e., moment-agnostic causal graph $G$) is also a DAG (see Corollary A.1 for details), and this DAG condition is standard in existing work (Strobl and Lasko, 2023; Yin et al., 2024). To ensure it, if mean causal graph $G^M$ contains $X_i \to X_j$, then variance causal graph $G^V$ cannot have the reverse edge $X_i \leftarrow X_j$ and must either (i) also include $X_i \to X_j$ or (ii) contain no edge. This means that the structural differences between $G^M$ and $G^V$ can only lie in the **presence or absence** of each edge.

By restricting possible structures in this way, we derive the sufficient (but not necessary) conditions:

**Theorem 3.5:** *Under Assumptions 3.1, 3.2, 3.3, and 3.4, mean and variance causal graphs $G^M$ and $G^V$ are identifiable from observational distribution $\mathrm{P}(\boldsymbol{X})$ if for $j = 1, \ldots, d$, (A) $m_j$ is a nonlinear function, (B) $v_j$ is a piecewise function, **but not a constant function**, and (C) $E_j$ is a Gaussian noise.*

*Proof sketch.* Our proof in Appendix B takes two steps. First, we show that Conditions (A), (B), and (C) ensure identifiability of the moment-agnostic causal graph $G$ for our mean-variance HNM. To do so, we prove that these conditions are sufficient to rule out the non-identifiable cases established by Khemakhem et al. (2021) for the original HNM, which also apply to our mean-variance HNM once the structural differences between $G^M$ and $G^V$ are restricted as above (Appendix B.1). Second, for $j \in \{1, \ldots, d\}$, we show that the parent set $\boldsymbol{X}_{\mathrm{pa}(j)}$ in $G$ can be decomposed into subsets $\boldsymbol{X}_{\mathrm{pa}^M(j)}$ and $\boldsymbol{X}_{\mathrm{pa}^V(j)}$ under Gaussian noise $E_j$. To this end, we prove that $\mathrm{P}(\boldsymbol{X})$ cannot coincide under different choices of $\boldsymbol{X}_{\mathrm{pa}^M(j)}$ and $\boldsymbol{X}_{\mathrm{pa}^V(j)}$, due to the non-constancy of $m_j$ and $v_j$ (Appendix B.2). □

*Remark* 3.6. Compared with Yin et al. (2024, Theorem 2), our Condition (B) on variance function $v_j$ is stronger, as it is violated when $v_j$ is a constant. However, excluding constant functions is standard for guaranteeing the identifiability, as done for ANMs (Peters et al., 2014, Proposition 17). The non-constancy of $m_j$ and $v_j$ is crucial to distinguish the causes based on the mean and variance.

Notably, our identifiability conditions in Theorem 3.5 are expressed in terms of the forms of mean and variance functions $m_j$ and $v_j$, as well as the distribution of noise $E_j$, which **can be easily enforced** through likelihood modeling to ensure the model remains within an identifiable class.

## 4 PROPOSED METHOD

### 4.1 MODEL FORMULATIONS

We develop a Bayesian approach for inferring the posterior distribution over adjacency matrices $\mathbf{A}^M$ and $\mathbf{A}^V$ of the mean and variance causal graphs, given observational data $\mathcal{D}$:

$$\mathrm{P}(\mathbf{A}^M, \mathbf{A}^V \mid \mathcal{D}) \propto \mathrm{P}(\mathbf{A}^M, \mathbf{A}^V)\,\mathrm{P}(\mathcal{D} \mid \mathbf{A}^M, \mathbf{A}^V), \tag{4}$$

where $P(\mathbf{A}^M, \mathbf{A}^V)$ is a prior, and $P(\mathcal{D} \mid \mathbf{A}^M, \mathbf{A}^V)$ is a marginal likelihood, obtained by integrating out the likelihood model parameters $\Theta$ as $\int P(\Theta \mid \mathbf{A}^M, \mathbf{A}^V) P(\mathcal{D} \mid \Theta, \mathbf{A}^M, \mathbf{A}^V) d\Theta$. Unfortunately, tractable computation of this integral requires restrictive parameterizations, such as linear Gaussian models, which may lead to inaccurate inferences when the model assumptions are violated.

For this reason, we approximate the posterior with a variational distribution (with parameters $\Phi$) as $P(\mathbf{A}^M, \mathbf{A}^V \mid \mathcal{D}) \approx P_\Phi(\mathbf{A}^M, \mathbf{A}^V)$. Below we present this DAG distribution and the likelihood.

### 4.1.1 DAG Distribution Model

We formulate the theoretically aligned DAG distribution model $P_\Phi(\mathbf{A}^M, \mathbf{A}^V)$ by ensuring that $\mathbf{A}^M$ and $\mathbf{A}^V$ are the DAGs with a shared permutation (Assumptions 3.3 and 3.4).

To guarantee that adjacency matrices $\mathbf{A}^M$ and $\mathbf{A}^V$ represent such DAGs, we build on a well-known result in graph theory and decompose them using a shared permutation matrix $\mathbf{\Pi}$:

$$\mathbf{A}^M = \mathbf{\Pi}^\top \mathbf{U}^M \mathbf{\Pi}, \quad \mathbf{A}^V = \mathbf{\Pi}^\top \mathbf{U}^V \mathbf{\Pi}, \tag{5}$$

where $\mathbf{U}^M, \mathbf{U}^V \in \{0,1\}^{d \times d}$ are upper-triangular matrices, and $\mathbf{\Pi} \in \{0,1\}^{d \times d}$ is a permutation matrix, in which every row and every column has exactly one element of 1 with all others 0. Eq. (5) simply states that permuting the elements in an upper-triangular matrix leads to a DAG adjacency matrix, e.g., $U_{\pi(i),\pi(j)}^M = A_{i,j}^M$, where $\pi$ is a permutation that takes $\pi(i) = j$ if and only if $\Pi_{i,j} = 1$.

Using the DAG decomposition in (5), we consider a variational distribution based on the following factorization: $P(\mathbf{A}^M, \mathbf{A}^V) = \sum_{\mathbf{U}^M, \mathbf{U}^V, \mathbf{\Pi}} P(\mathbf{U}^M) P(\mathbf{U}^V) P(\mathbf{\Pi})$, where the summation is over all upper-triangular matrices $\mathbf{U}^M$ and $\mathbf{U}^V$ and permutation matrices $\mathbf{\Pi}$ that are compatible with a DAG pair, $\mathbf{A}^M$ and $\mathbf{A}^V$. Unfortunately, the exact computation of this summation is intractable, as the number of valid permutation matrices $\mathbf{\Pi}$ grows exponentially with the number of nodes $d$.

For this reason, we take a sampling-based learning approach by generalizing the idea of differentiable DAG sampling (DDS) (Charpentier et al., 2022) to our setup with two DAG adjacency matrices $\mathbf{A}^M$ and $\mathbf{A}^V$. To enable backpropagation in a backward pass, we approximately compute the gradient of the sampling operations for binary matrices $\mathbf{U}^M$, $\mathbf{U}^V$, and $\mathbf{\Pi}$ in (5) by leveraging their continuous relaxation $\tilde{\mathbf{U}}^M, \tilde{\mathbf{U}}^V \in [0,1]^{d \times d}$ and $\tilde{\mathbf{\Pi}} \in \mathbb{R}^{d \times d}$. To obtain such continuous matrices $\tilde{\mathbf{U}}^M$ and $\tilde{\mathbf{U}}^V$, we employ the *Gumbel-Softmax trick* (Jang et al., 2017), which allows differentiable approximation using i.i.d. standard Gumbel noises $g_0^M, g_0^V, g_1^M, g_1^V \sim \text{Gumbel}(0)$:

$$\tilde{U}_{i,j}^M = \frac{e^{(\log \phi_{i,j}^M + g_1^M)/\tau^M}}{e^{(\log \phi_{i,j}^M + g_0^M)/\tau^M} + e^{(\log(1-\phi_{i,j}^M) + g_1^M)/\tau^M}}, \tilde{U}_{i,j}^V = \frac{e^{(\log \phi_{i,j}^V + g_1^V)/\tau^V}}{e^{(\log \phi_{i,j}^V + g_0^V)/\tau^V} + e^{(\log(1-\phi_{i,j}^V) + g_1^V)/\tau^V}},$$

where $\phi_{i,j}^M, \phi_{i,j}^V \in [0,1]$ denote the Bernoulli distribution parameters, and $\tau^M, \tau^V \geq 0$ are the hyperparameters that control the smoothness of the distribution. To compute $\tilde{\mathbf{\Pi}}$, we directly adopt the DDS's sampling scheme, which approximates the sorting order of perturbed log probabilities:

$$\tilde{\mathbf{\Pi}} = \text{SoftSort}(\log \boldsymbol{\psi} + \mathbf{g}), \tag{6}$$

where $\boldsymbol{\psi} \in \mathbb{R}^d$ denotes the categorical distribution parameters, $\mathbf{g} \in \mathbb{R}^d$ is a vector of i.i.d. standard Gumbel noise, and SoftSort is a differentiable approximation to a sorting operation, which returns a continuous relaxation of permutation matrix representing the sorted order of the input elements. We detail the formulation of SoftSort function in Appendix C.2.

In a forward pass, we sample binary matrices $\mathbf{U}^M, \mathbf{U}^V, \mathbf{\Pi} \in \{0,1\}^{d \times d}$ by applying the argmax operator on $\tilde{\mathbf{U}}^M$ and $\tilde{\mathbf{U}}^V$ (e.g., $U_{i,j}^M = \arg\max[1 - \tilde{U}_{i,j}^M, \tilde{U}_{i,j}^M]$) and a row-wise argmax operation on $\tilde{\mathbf{\Pi}} \in \mathbb{R}^{d \times d}$. By plugging the sample of $\mathbf{U}^M, \mathbf{U}^V$, and $\mathbf{\Pi}$ into Eq. (5), we sample DAG adjacency matrices $\mathbf{A}^M, \mathbf{A}^V \sim P_\Phi(\mathbf{A}^M, \mathbf{A}^V)$, where $\Phi = \{\boldsymbol{\phi}^M, \boldsymbol{\phi}^V, \boldsymbol{\psi}\}$ is a set of parameters.

### 4.1.2 Likelihood Model

Using DAG sample $\mathbf{A}^M$ and $\mathbf{A}^V$, we model a likelihood based on our mean-variance HNM in (3).

For each $j = 1, \ldots, d$, we parameterize mean and variance functions $m_j$ and $v_j$ using multi-layer perceptrons (MLPs) with parameters $\boldsymbol{\theta}_j^M$ and $\boldsymbol{\theta}_j^V$. To represent the (possibly different) inputs of

these MLPs, we apply masking to $X$ using the $j$-th column vectors in $\mathbf{A}^M$ and $\mathbf{A}^V$, denoted by $\boldsymbol{A}_j^M, \boldsymbol{A}_j^V \in \{0, 1\}^d$, leading to the following structural equation parameterization:

$$X_j = m_j(\boldsymbol{A}_j^M \odot \boldsymbol{X}; \boldsymbol{\theta}_j^M) + v_j(\boldsymbol{A}_j^V \odot \boldsymbol{X}; \boldsymbol{\theta}_j^V)E_j \quad \text{for } j = 1, \dots, d, \tag{7}$$

where $\odot$ denotes the Hadamard product. By assuming, without loss of generality, that noise $E_j$ follows a standard Gaussian distribution $\mathcal{N}(0, 1)$, we parameterize the conditional distribution of $X_j$ in (7) as multivariate Gaussian $\mathcal{N}(m_j(\boldsymbol{A}_j^M \odot \boldsymbol{X}; \boldsymbol{\theta}_j^M), (v_j(\boldsymbol{A}_j^V \odot \boldsymbol{X}; \boldsymbol{\theta}_j^V))^2)$.

To satisfy Conditions (A) and (B) in Theorem 3.5, we formulate all MLPs in our experiments using leaky rectified linear unit (ReLU) activations, which are piecewise nonlinear and cannot be a constant. To ensure that the variance function value is strictly positive (i.e., $v_j(\cdot) > 0$), we model $\log v_j(\cdot)$ using an MLP and apply the exponential function to it as a final transformation.

### 4.2 Model Parameter Learning

To obtain approximated posterior $P_\Phi(\mathbf{A}^M, \mathbf{A}^V) \simeq P(\mathbf{A}^M, \mathbf{A}^V \mid \mathcal{D})$, we learn parameters $\Phi = \{\boldsymbol{\phi}^M, \boldsymbol{\phi}^V, \boldsymbol{\psi}\}$, and $\Theta = \{(\boldsymbol{\theta}_j^M, \boldsymbol{\theta}_j^V)_{j=1}^d\}$, such that the Kullback Leibler (KL) divergence between variational distribution $P_\Phi(\mathbf{A}^M, \mathbf{A}^V)$ and true posterior $P(\mathbf{A}^M, \mathbf{A}^V \mid \mathcal{D})$ is minimized. Since minimizing this KL divergence is equivalent to maximizing the evidence lower bound (ELBO), we aim to maximize the following ELBO-based objective function:

$$\max_{\Phi, \Theta} \quad \mathbb{E}_{\mathbf{A}^M, \mathbf{A}^V \sim P_\Phi} \left[ \log P_\Theta(\mathcal{D} \mid \mathbf{A}^M, \mathbf{A}^V) \right] - \lambda \Omega_{\Phi, \Theta}, \tag{8}$$

where $\log P_\Theta(\mathcal{D} | \mathbf{A}^M, \mathbf{A}^V)$ is a log likelihood, $\lambda \geq 0$ is a parameter for the sparsity regularizer:

$$\Omega_{\Phi, \Theta} = \lambda_\Phi \text{KL} \left( P_\Phi(\mathbf{U}^M, \mathbf{U}^V) \,\|\, P(\mathbf{U}^M, \mathbf{U}^V) \right) + \lambda_{\Theta^M} \sum_{j=1}^d \|\boldsymbol{\theta}_j^M\|_2^2 + \lambda_{\Theta^V} \sum_{j=1}^d \|\boldsymbol{\theta}_j^V\|_2^2, \tag{9}$$

where $\lambda_\Phi, \lambda_{\Theta^M}, \lambda_{\Theta^V} \geq 0$ are hyperparameters. Following DDS (Charpentier et al., 2022), we do not put any prior on permutation matrix $\boldsymbol{\Pi}$ in Eq. (9) to retain tractable computation. We encourage sparsity in the edge presence probabilities by setting prior probabilities $P(\mathbf{U}^M, \mathbf{U}^V)$ to small values.

To solve the optimization problem in (8), we repeat three steps. First, we sample DAG adjacency matrices $\mathbf{A}^M$ and $\mathbf{A}^V$ from $P_\Phi(\mathbf{A}^M, \mathbf{A}^V)$. Then we use this single DAG sample to approximate the expectation in Eq. (8) as the Gaussian log likelihood for observational dataset $\mathcal{D} = \{\boldsymbol{x}_1, \dots, \boldsymbol{x}_n\}$:

$$\log P_\Theta(\mathcal{D} | \mathbf{A}^M, \mathbf{A}^V) = -\frac{1}{n} \sum_{i=1}^n \sum_{j=1}^d \left( \frac{(x_{i,j} - m_j(\boldsymbol{x}_{i,\text{pa}^M(j)}; \boldsymbol{\theta}_j^M))^2}{2(v_j(\boldsymbol{x}_{i,\text{pa}^V(j)}; \boldsymbol{\theta}_j^V))^2} + \log v_j(\boldsymbol{x}_{i,\text{pa}^V(j)}; \boldsymbol{\theta}_j^V) \right),$$

where $\boldsymbol{x}_{i,\text{pa}^M(j)} := \boldsymbol{A}_j^M \odot \boldsymbol{x}_i$ and $\boldsymbol{x}_{i,\text{pa}^V(j)} := \boldsymbol{A}_j^V \odot \boldsymbol{x}_i$ are the masked input vectors obtained from observation $\boldsymbol{x}_i$. Finally, we perform a gradient-based update for $\Phi$ and $\Theta$ based on Eq. (8).

To effectively perform such a gradient-based optimization, we face two challenges.

**1. Difficulty in Maximum Likelihood Estimation (MLE):** Maximizing the likelihood with respect to parameters $\Theta$ is *ill-posed* in the sense that it can be arbitrarily increased by inflating the variance function value rather than by reducing the residual errors (Seitzer et al., 2022). Moreover, gradient-based optimization slows around the data points with high true variance because the gradients with respect to $m_j$ and $v_j$ scale inversely with variance function $v_j(\boldsymbol{x}_{i,\text{pa}^V(j)}; \boldsymbol{\theta}_j^V)$ (Skafte et al., 2019).

We overcome these two issues by building on recent advances in heteroscedastic noise regression (Stirn et al., 2023; Wong-Toi et al., 2024). To address the ill-posedness of MLE, we impose $l_2$ regularization on $\Theta$, as shown in Eq. (9). To resolve the optimization inefficiency, we alternately take two steps. First, we update all the parameters except for the variance-related ones, i.e., $\Theta \backslash \{(\boldsymbol{\theta}_j^V)_{j=1}^d\}$ and $\Phi \backslash \boldsymbol{\phi}^V$, using the gradient of mean squared error (MSE), $\frac{1}{n} \sum_{i,j} (x_{i,j} - m_j(\boldsymbol{x}_{i,\text{pa}^M(j)}; \boldsymbol{\theta}_j^M))^2$, which is equivalent to scaling the standard gradient by $(v_j(\boldsymbol{x}_{i,\text{pa}^V(j)}; \boldsymbol{\theta}_j^V))^2$. This scaling approximates a computationally demanding second-order Newton step with an inverse Jacobian, thus enabling an efficient, curvature-aware optimization. Second, we update variance-related parameters $\{(\boldsymbol{\theta}_j^V)_{j=1}^d\}$ and $\boldsymbol{\phi}^V$ using the standard gradient. Algorithm 1 summarizes the overall procedure. The total time complexity is dominated by likelihood evaluation and is identical to Yin et al. (2024).

---

**Algorithm 1** Parameter Learning for Mean and Variance Causal Graphs

---

**Require:** Observational dataset $\mathcal{D} = \{\boldsymbol{x}_1, \ldots, \boldsymbol{x}_n\}$; parameters $\Phi, \Theta$; node-ordering constraints $\mathcal{O}$
 1: Initialize all parameters $\Phi = \{\boldsymbol{\phi}^M, \boldsymbol{\phi}^V, \boldsymbol{\psi}\}$ and $\Theta = \{(\boldsymbol{\theta}_j^M, \boldsymbol{\theta}_j^V)_{j=1}^d\}$
 2: **while** Stopping criterion not met **do**
 3:     **while** Updates for parameters $\Theta \setminus \{(\boldsymbol{\theta}_j^V)_{j=1}^d\}$ and $\Phi \setminus \{\boldsymbol{\phi}^V\}$ not converged **do**
 4:         Sample mean and variance DAGs $\mathbf{A}^M, \mathbf{A}^V \sim \mathrm{P}_\Phi(\mathbf{A}^M, \mathbf{A}^V)$
 5:         Compute ELBO-based objective function in Eq. (8)
 6:         Update parameters $\Theta \setminus \{(\boldsymbol{\theta}_j^V)_{j=1}^d\}$ and $\Phi \setminus \{\boldsymbol{\phi}^V\}$ using scaled gradient
 7:         **if** node-ordering constraints $\mathcal{O}$ are provided **then**
 8:             Project updated parameters $\boldsymbol{\psi}$ onto the feasible set by Eq. (11)
 9:         **end if**
10:     **end while**
11:     **while** Updates for variance-related parameters $\{(\boldsymbol{\theta}_j^V)_{j=1}^d\}$ and $\{\boldsymbol{\phi}^V\}$ not converged **do**
12:         Sample mean and variance DAGs $\mathbf{A}^M, \mathbf{A}^V \sim \mathrm{P}_\Phi(\mathbf{A}^M, \mathbf{A}^V)$
13:         Compute ELBO-based objective function in Eq. (8)
14:         Update variance-related parameters $\{(\boldsymbol{\theta}_j^V)_{j=1}^d\}$ and $\{\boldsymbol{\phi}^V\}$ using standard gradient
15:     **end while**
16: **end while**
17: **return** Learned parameters $\Phi$ and $\Theta$

---

**2. Large Parameter Search Space:** Fitting our DAG distribution model parameter $\Phi$ for the two causal DAGs is challenging because it is inevitably more high-dimensional than the existing models for a single (moment-agnostic) causal DAG. Although we effectively reduce the number of parameters by sharing permutation distribution parameter $\boldsymbol{\psi} \subset \Phi$ (Section 4.1.1), its estimation might remain difficult under small sample-size scenarios. To tackle this challenge, we develop an approach that reduces the search space of $\boldsymbol{\psi}$ by incorporating prior knowledge about the node orderings.

### 4.3 PRIOR KNOWLEDGE INCORPORATION

Incorporating domain knowledge about the ground-truth graph has been a key strategy for addressing small-sample setups (Niinimäki et al., 2016; Oyen et al., 2017; Li and Beek, 2018; Hasan and Gani, 2022). In this work, we focus on pairwise node-ordering knowledge—for example, knowing that $X_i$ is an upstream regulator of gene $X_j$—which is often available in practice (Ban et al., 2024).

Suppose that we have access to a set of pairwise node-ordering constraints $\mathcal{O} = \{\pi(i) < \pi(j) \mid (i,j) \in \mathcal{S}\}$ for a set of variable index pairs $\mathcal{S} \subseteq \{(i,j) \mid i,j \in \{1,\ldots,d\}\}$, where $\pi(i) < \pi(j)$ denotes a pairwise node-ordering constraint that variable $X_i$ precedes $X_j$ in permutation $\pi$.

Then we can easily incorporate such node-ordering constraints, thanks to the sorting-based sampling scheme in (6). Since this scheme samples the continuous relaxation of the permutation matrix by computing a sorting order for parameter vector $\boldsymbol{\psi} \in \mathbb{R}^d$, we can impose the node-ordering constraints in $\mathcal{O}$ by forcing the parameter values to satisfy the following inequality constraint set:

$$\mathcal{I}(\boldsymbol{\psi}) = \{\psi_i + c_{i,j} \leq \psi_j \mid (i,j) \in \mathcal{S}\}, \tag{10}$$

where $c_{i,j} > 0$ is a hyperparameter that ensures $\psi_i < \psi_j$. To satisfy these constraints, we project the updated parameter values $\boldsymbol{\psi}'$ onto the feasible set by solving the constrained least squares:

$$\boldsymbol{\psi}_{\text{new}} = \arg\min_{\boldsymbol{\rho} \in \mathcal{I}(\boldsymbol{\rho})} \|\boldsymbol{\rho} - \boldsymbol{\psi}'\|_2^2, \tag{11}$$

which can be efficiently solved with a quadratic programming (QP) solver. The constraints in (11) are soft constraints, since they might not hold in sampled permutations due to the perturbation by Gumbel noise in Eq. (6). Hyperparameter $c_{i,j}$ strikes a trade-off between the robustness to this noise and possible parameter values; we set $c_{i,j} = 1.5$ for all $i,j \in \{1,\ldots,d\}$ in our experiments.

## 5 EXPERIMENTS

**Baselines:** We compare our method with four baselines. (i) Two Bayesian methods based on ANM likelihoods: Metropolis-coupled Markov chain Monte Carlo (**MC3**) (Giudici and Castelo, 2003),

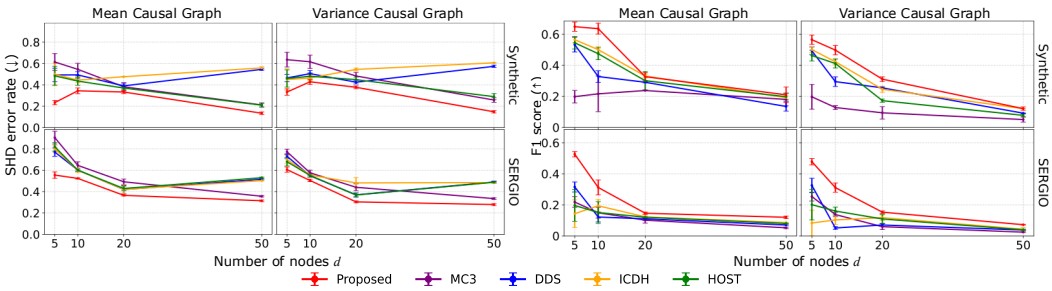

Figure 2: Mean and variance causal graph inference performance on synthetic and SERGIO datasets with sample size $n = 500$. Achieving **both** lower SHD rate (left) and higher F1 score (right) is better.

Table 1: Results on Sachs dataset. $\downarrow$ and $\uparrow$ denote "lower is better" and "higher is better".

|  | $n = 100$ | | $n = 200$ | | $n = 853$ | |
|---|---|---|---|---|---|---|
|  | SHD ($\downarrow$) | F1 ($\uparrow$) | SHD ($\downarrow$) | F1 ($\uparrow$) | SHD ($\downarrow$) | F1 ($\uparrow$) |
| MC3 | $22.6 \pm 1.1$ | $0.19 \pm 0.04$ | $20.3 \pm 1.3$ | $0.20 \pm 0.07$ | $18.9 \pm 0.7$ | $0.14 \pm 0.05$ |
| DDS | $17.6 \pm 0.7$ | $0.20 \pm 0.03$ | $16.6 \pm 0.6$ | $0.20 \pm 0.02$ | $15.0 \pm 0.7$ | $0.28 \pm 0.02$ |
| ICDH | $20.0 \pm 1.0$ | $\mathbf{0.21 \pm 0.01}$ | $17.5 \pm 0.5$ | $0.22 \pm 0.01$ | $14.5 \pm 0.5$ | $0.27 \pm 0.03$ |
| HOST | $16.1 \pm 0.5$ | $0.19 \pm 0.01$ | $15.0 \pm 0.3$ | $\mathbf{0.33 \pm 0.02}$ | $13.5 \pm 0.8$ | $\mathbf{0.38 \pm 0.03}$ |
| PROPOSED | $\mathbf{16.0 \pm 0.3}$ | $0.20 \pm 0.02$ | $\mathbf{14.9 \pm 0.4}$ | $0.31 \pm 0.02$ | $13.7 \pm 0.5$ | $0.36 \pm 0.02$ |
| PROPOSED +25% | $14.9 \pm 0.4$ | $0.34 \pm 0.03$ | $14.8 \pm 0.5$ | $0.35 \pm 0.02$ | $13.4 \pm 0.4$ | $0.37 \pm 0.02$ |
| PROPOSED +50% | $\mathbf{13.2 \pm 0.5}$ | $\mathbf{0.36 \pm 0.02}$ | $\mathbf{13.2 \pm 0.3}$ | $\mathbf{0.36 \pm 0.02}$ | $\mathbf{13.1 \pm 0.2}$ | $\mathbf{0.45 \pm 0.03}$ |

a traditional posterior-sampling approach using linear-Gaussian models; and a variational inference method, **DDS** (Charpentier et al., 2022). (ii) Two point-estimation methods based on HNM likelihoods, identifiable causal discovery under heteroscedastic data (**ICDH**) (Yin et al., 2024) and heteroscedastic causal structure learning (**HOST**) (Duong and Nguyen, 2023). Following Yin et al. (2024), we exclude VarSort (Reisach et al., 2021), as its performance is highly unstable under heteroscedastic noise, due to the reliance on marginal variances (see Appendix E.5 for details).

To make a fair comparison, we test **both** moment-specific and moment-agnostic graph inference performance, mirroring Yin et al. (2024), who evaluated **ICDH** on both ANM and HNM datasets.

## 5.1 MEAN AND VARIANCE CAUSAL GRAPH INFERENCE FROM SIMULATED DATA

**Data:** We evaluate our method on two types of datasets. One is a synthetic dataset generated from mean-variance HNMs, where both the mean and variance functions are modeled by MLPs with randomly initialized parameters. The other is a semi-synthetic dataset generated by SERGIO (Dibaeinia and Sinha, 2020), a gene expression simulator that produces *realistic* single-cell transcriptomic data using the parameters tuned to real data. We randomly sample ground truth causal graphs for each type of datasets from the Erdős–Rényi (ER) and the scale-free (SF) models (Appendix D.3).

**Evaluation Metrics:** We use the structural Hamming distance (SHD) and F1 score (Appendix D.5). We report the mean and standard deviation (SD) over 20 randomly generated datasets.

**Performance comparison:** Figure 2 presents the SHD rate (i.e., the SHD divided by its maximum possible value, $\mathrm{SHD}/\binom{d}{2}$) and the F1 score. Our method outperforms all baselines on both synthetic and semi-synthetic datasets, demonstrating its effectiveness for moment-driven causal discovery.

On synthetic datasets (top row of Figure 2), **MC3** and **DDS** yield poor F1 scores because their ANM-based likelihoods are misspecified under heteroscedasticity. **ICDH** and **HOST** perform worse at inferring the variance graph, suggesting that their HNM-based inference is driven by the underlying mean-graph structure and therefore overlooks the drivers of heteroscedasticity. These findings highlight the importance of moment-driven causal discovery for understanding complex mechanisms.

On semi-synthetic datasets (bottom row of Figure 2), which reflect the complex nonlinearity and heteroscedasticity of gene expression data, our method achieves substantial gains over the base-

lines—especially in the F1 score—underscoring its practical reliability and versatility. Although our method also performs best at $d = 50$, accurate inference in such high-dimensional settings remains challenging due to the accumulation of heteroscedastic noise. Incorporating more advanced heteroscedastic noise regression techniques to tackle this challenge is left as our future work.

## 5.2 REAL-WORLD DATA EXPERIMENTS

**Data:** We employ a well-established benchmark dataset, called the Sachs dataset (Sachs et al., 2005), which contains 853 observations of the expression levels of $d = 11$ proteins in human cells. We test each method under $n = 100, 200$, and 853 observations with 20 random restarts.

**Moment-Agnostic Graph Inference Performance:** As the ground-truth mean and variance graph structures are unknown, we evaluate the moment-agnostic causal graph inference performance.

Table 1 presents the results. Although our method is not tailored for moment-agnostic graph inference and learns additional parameters for inferring two separate causal graphs, it achieves comparable performance to the state-of-the-art **HOST** method. When it leverages randomly selected 25% and 50% of pairwise node orderings, the performance improves with an increase in their number, especially for small sample-size setups ($n = 100$ and 200), underscoring its practical utility.

**Case Study on Variance Edge Detection:** Guided by prior biological evidence (Filippi et al., 2016), we hypothesize that the MEK $\rightarrow$ ERK edge in the ground-truth protein regulatory network represents a causal relationship at the variance level, and we assess whether our method can identify this edge.

Consistent with this hypothesis, from the datasets with sample sizes $n = 100, 200$ and 853, our method successfully assigns high posterior probabilities of 0.585, 0.592, and 0.620, respectively, for MEK $\rightarrow$ ERK in the inferred variance graph structure (see Appendix E.3 for details). These results underscore the effectiveness of our Bayesian, moment-driven approach in discovering the drivers of heteroscedasticity under data scarcity settings in practical real-world applications.

## 5.3 ADDITIONAL EXPERIMENTS

To further evaluate the performance of our method, we conduct additional experiments. Below we summarize the key findings of the results, which are detailed in Appendix E:

- **Dense mean and variance graphs:** Our method works best in a dense setting (Appendix E.1).
- **Moment-agnostic graph inference from ANM and HNM datasets:** On these homoscedastic and heteroscedastic datasets, our method is comparable to the ANM-based and HNM-based baselines, demonstrating its robustness in moment-agnostic graph inference (Appendix E.2).
- **Run time comparison:** Our method takes twice as long as **DDS**, as it infers two separate causal graphs unlike this baseline. Our prior knowledge incorporation framework adds only modest overhead to the run time, making it practical for achieving high performance (Appendix E.4).

## 6 DISCUSSION

**Limitations and Future Work:** Despite its theoretically grounded foundation, our method relies on the Gaussian noise assumption to ensure the separability of moment-agnostic causal graph into mean and variance causal graphs. While we observe its stronger moment-agnostic graph inference performance than HNM-based baselines on non-Gaussian synthetic datasets (see Appendix E.6 for details), establishing a general identifiability theory beyond Gaussian noise remains open and will likely require the identification of *higher-order moment causal graphs*, which would elucidate how distributional characteristics like skewness and kurtosis are determined in real-world phenomena. We view this challenging yet important problem as a promising direction for future work.

**Conclusion:** This paper is the first to propose moment-driven causal discovery, advancing causal discovery by enabling moment-specific reasoning about complex causal mechanisms. As a first step, we propose an efficient variational inference framework that directly learns the mean and variance causal graphs, without relying on any moment-agnostic graph inference results. It can further improve sample efficiency by incorporating prior knowledge, and provides principled uncertainty quantification for the inferred graphs, grounded in theoretical identifiability guarantees.

## ETHICS STATEMENT AND REPRODUCIBILITY

*Ethics Statement*: We made every effort to ensure a fair performance comparison despite differences in inference targets across methods (i.e., our method infers mean and variance causal graphs, whereas the baselines infer moment-agnostic causal graphs). To this end, we thoroughly evaluated all methods on **both** inference tasks and reported results accordingly in the main text and supplementary materials. As described in Section 5, this evaluation is based on the same spirit as Yin et al. (2024), who extensively evaluated their HNM-based method on both ANM and HNM datasets.

Our framework poses no direct harm by itself. As with standard causal discovery methods, however, misuse may lead to unintended negative consequences in downstream tasks, such as decision-making based on treatment effect estimation (Shalit et al., 2017; Emezue et al.; Chikahara and Ushiyama, 2024; Moreira et al., 2025). A notable downstream application is algorithmic fairness (Kusner et al., 2017; Chiappa, 2019; Li et al., 2024; Zuo et al., 2024). When using inferred causal graph structures to identify latent sensitive attributes (see Section 1), additional safeguards (e.g., domain-expert supervision) may be necessary to mitigate potential bias in algorithmic decisions.

*Reproducibility*: We describe the experimental settings in Appendix D. We tune hyperparameters with the publicly available tool (Akiba et al., 2019) using a fixed random seed, as detailed in Appendix D.2. **Our source code and datasets are available** at the following anonymized link: `https://osf.io/download/68cfd10f5c49af5a1c413fd9/?view_only=a5ed210a1b7545fe835c8370e1feee38` and will be made public upon publication.

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

## A  ASSUMPTIONS

This section describes the assumptions for deriving the identifiability conditions in Appendix B.

We make standard assumption in the literature on causal discovery:

**Assumption 3.1** (Causal sufficiency)**:** *Exogenous noises satisfy $E_i \perp\!\!\!\perp E_j$ for any $i, j \in \{1, \ldots, d\}$.*

Assumption 3.1 requires that there is no unobserved common cause of observed variables $\boldsymbol{X}$; this assumption is widely used in the causal inference literature (Glymour et al., 2019). Satisfying Assumption 3.1 and the DAG condition on $G$ ensures the causal Markov condition (Pearl, 2009), which assumes that each variable is conditionally independent of the variables represented as its non-descendants in $G$.

In addition, we make another classical assumption that ensures identifiability:

**Assumption 3.2** (Causal minimality)**:** *Joint distribution $\mathrm{P}(\boldsymbol{X})$ satisfies the causal minimality condition with respect to moment-agnostic causal graph $G$.*

Causal minimality requires that joint distribution $\mathrm{P}(\boldsymbol{X})$ be Markov with respect to (moment-agnostic) causal graph $G$ but not with respect to any proper subgraph of $G$ (Spirtes et al., 2001). Roughly speaking, satisfying it with respect to causal graph $G$ implies that removing any edge from $G$ will violate the causal Markov condition, and hence the conditional independence relations in joint distribution $\mathrm{P}(\boldsymbol{X})$ will no longer be compatible with $G$. Causal minimality, which is standard for deriving the identifiability under a restricted class of SCMs, is weaker than the faithfulness assumption (Peters et al., 2014).

Furthermore, we make two additional assumptions that are specific to our results:

**Assumption 3.3:** *Mean and variance graphs $G^M$ and $G^V$ are directed acyclic graphs (DAGs).*

**Assumption 3.4** (Shared permutation condition)**:** *There exists an identical permutation (a.k.a., topological ordering) of mean and variance causal graphs $G^M$ and $G^V$.*

As mentioned in Section 3.2, both assumptions are needed to ensure that moment-agnostic causal graph $G$ is a DAG, which has the same permutation with $G^M$ and $G^V$:

**Corollary A.1.** *Under Assumptions 3.3 and 3.4, moment-agnostic causal graph $G$ is a DAG with the same permutation as the mean and variance causal graphs $G^M$ and $G^V$.*

*Proof.* Let $\pi$ be a permutation (i.e., a valid topological ordering) of both $G^M$ and $G^V$. We first show that $\pi$ is also a valid topological ordering for their union graph $G$ and then prove that $G$ is a DAG.

Since a directed path in a DAG is a concatenation of a finite number of directed edges, it is sufficient to verify that $\pi(u) < \pi(v)$ holds for every edge $(u, v) \in G$. By assumption, $\pi$ is a valid topological ordering over both $G^M$ and $G^V$. Thus, for every edge $(u, v) \in G$, regardless of whether it is present in $G^M$, $G^V$, or both, we have $\pi(u) < \pi(v)$. Therefore, $\pi$ is a valid topological ordering for $G$.

Next, we show that the union graph $G$ is acyclic. Suppose that $G$ contains a directed cycle $v_1 \rightarrow \cdots \rightarrow v_k \rightarrow v_1$ over the node(s) $v_1, \ldots, v_k$ ($k \geq 1$). Then, since $\pi$ is a topological ordering of $G$, it must satisfy $\pi(v_1) < \cdots < \pi(v_k) < \pi(v_1)$; however, $\pi(v_1) < \pi(v_1)$ is a contradiction.

Therefore, such a cycle cannot exist, and the union graph $G$ must be acyclic. Hence, $G$ is a DAG and $\pi$ is a valid topological ordering for $G$. $\qquad\square$

*Remark* A.2. The DAG condition on the causal graph (Corollary A.1) is a standard assumption for showing the identifiability in the HNM literature (Xu et al., 2022; Strobl and Lasko, 2023; Lin et al., 2025; Yin et al., 2024). Accordingly, the corresponding Assumptions 3.3 and 3.4 in our setup are also standard.

## B  IDENTIFIABILITY PROOF

This section presents the proof of Theorem 3.5. To prove it, we take three steps:

(i) We show the identifiability of moment-agnostic causal graph $G$ of our mean-variance HNM for bivariate case $d = 2$ (Appendix B.1.1).

(ii) We extend these results to a multivariate setup $d > 2$ (Appendix B.1.2).

(iii) We prove Theorem 3.5 by deriving the sufficient conditions for the identification of mean and variance causal graphs $G^M$ and $G^V$ (Appendix B.2).

## B.1 IDENTIFIABILITY OF MOMENT-AGNOSTIC CAUSAL GRAPH OF MEAN-VARIANCE HNM

### B.1.1 BIVARIATE CASE ($d = 2$)

We prove bivariate identifiability of a moment-agnostic causal graph $G$ in two steps:

(i-1) We illustrate *unidentifiable scenarios*, where the causal direction between $X_1$ and $X_2$ in moment-agnostic causal graph $G$ is unidentifiable from joint distribution $\mathrm{P}(X_1, X_2)$. To derive such scenarios, we apply the results of Khemakhem et al. (2021), which also hold for our mean-variance HNM under our assumptions.

(i-2) We derive the sufficient conditions that exclude these unidentifiable scenarios.

To highlight the differences between the original HNM and our mean-variance HNM, below we overview the existing results on the original HNM and then extend them to our mean-variance HNM.

**Results on the original HNM:** Khemakhem et al. (2021) show that two (original) HNMs with opposite causal directions can induce the same distribution $\mathrm{P}(X_1, X_2)$ under the following two scenarios:

**Theorem B.1** (Khemakhem et al. (2021, Theorem 2))**:** *Let $E_1, E_2 \sim \mathcal{N}(0, 1)$ be standard Gaussian noises, and let $m_1, m_2, v_1$ and $v_2$ be the twice-differentiable scalar functions on $\mathbb{R}$ that satisfy $v_1(\cdot) > 0$ and $v_2(\cdot) > 0$. Assume that $X_2$'s values are given by the forward model:*

$$X_2 = m_2(X_1) + v_2(X_1)E_2, \qquad (12)$$

*where $E_2 \perp\!\!\!\perp X_1$. Suppose that joint distribution $\mathrm{P}(X_1, X_2)$ is also compatible with the backward model:*

$$X_1 = m_1(X_2) + v_1(X_2)E_1, \qquad (13)$$

*where $E_1 \perp\!\!\!\perp X_2$. Then either of the two scenarios must hold:*

1. *$(m_2, v_2) = \left(\frac{B}{H}, \frac{1}{H}\right)$ and $(m_1, v_1) = \left(\frac{B'}{H'}, \frac{1}{H'}\right)$ where $B$ and $B'$ are polynomials of degree two or less, $H > 0$ and $H' > 0$ are two-order polynomials, and $\mathrm{P}(X_1), \mathrm{P}(X_2)$ are strictly log-mix-rational-log.[1]*

2. *$m_1, m_2$ are linear, $v_1, v_2$ are constant, and $\mathrm{P}(X_1), \mathrm{P}(X_2)$ are Gaussian distribution.*

Assuming that $E_1$ and $E_2$ are (standard) Gaussian noises, Theorem B.1 shows that the moment-agnostic causal graph becomes unidentifiable only in Scenarios 1 and 2. While function $v_j$ ($j = 1, 2$) is a constant under scenario 2, it **cannot** be a constant under scenario 1 because it must be the inverse of **two-order** polynomial function.

Building on this result, Yin et al. (2024) provide the identifiability conditions for the original HNM:

**Theorem B.2** (Yin et al. (2024, Theorem 2 (for $d = 2$)))**:** *Under Assumptions 3.1, 3.2 and the condition that the (moment-agnostic) causal graph $G$ is a DAG, $G$ is identifiable from observational data distribution $\mathrm{P}(X_1, X_2)$ if $\mathrm{P}(X_1, X_2)$ is generated from the (original) HNM in (2) that satisfies the following conditions for each $j = 1, 2$: (a) $m_j$ is a nonlinear function, (b) $v_j$ is a piecewise function, and (c) $E_j$ is Gaussian noise.*

Condition (c) is necessary to ensure the Gaussian noise assumption in Theorem B.1. Conditions (a) and (b) suffice to rule out Scenarios 2 and 1, respectively: Nonlinear $m_j$ excludes the linearity required in Scenario 2, and piecewise $v_j$ cannot be expressed as a polynomial form in Scenario 1.

**Extension to mean-variance HNM:** We now extend Theorem B.2 to our mean-variance HNM by taking into account the difference between the original and our mean-variance HNMs.

---

[1]See Khemakhem et al. (2021, Definition 1) for the definition of the (strictly) log-mix-rational-log density.

As discussed in Section 3.1, any instance of the mean-variance HNM can be reformulated as the original HNM by incorporating input masking into the functions in the original HNM. To avoid confusion, we use the notation $m_j^O(\boldsymbol{X}_{\mathrm{pa}(j)})$ and $v_j^O(\boldsymbol{X}_{\mathrm{pa}(j)})$ ($j = 1, 2$) for these functions in this section. Using this notation, the structural equation of $X_2$ when $X_1 \to X_2$ can be formulated as

$$X_2 = m_2(B_{\mathrm{pa}^M(2)}^M X_1) + v_2(B_{\mathrm{pa}^V(2)}^V X_1)E_2 = m_2^O(X_1) + v_2^O(X_1)E_2,$$

where $B_{\mathrm{pa}^M(2)}^M, B_{\mathrm{pa}^V(2)}^V \in \{0, 1\}$ are binary masking variables; for instance, if $B_{\mathrm{pa}^M(2)}^M = 0$, then function $m_2^O(X_1)$ becomes constant with respect to $X_1$. Note that these masking variables, $B_{\mathrm{pa}^M(2)}^M$ and $B_{\mathrm{pa}^V(2)}^V$, are assumed to be not simultaneously 0 under causal minimality (Assumption 3.2), ensuring that at least either of $m_2^O(X_1)$ and $v_2^O(X_1)$ depends on $X_1$.

Fortunately, Theorem B.1 already covers the cases where the functions are constants, as it only assumes their twice-differentiability, which is trivially satisfied by constant functions. Therefore, the unidentifiable scenarios for our mean-variance HNM are again limited to Scenarios 1 and 2. We show that Conditions (a), (b), and (c) in Theorem B.2 are sufficient to exclude these scenarios:

**Theorem B.3:** *Under Assumptions 3.1, 3.2, 3.3, and 3.4, moment-agnostic causal graph $G$ of the mean-variance HNM is identifiable from distribution $\mathrm{P}(X_1, X_2)$ if for $j = 1, 2$, (a) $m_j$ is a nonlinear function, (b) $v_j$ is a piecewise function, and (c) $E_j$ is Gaussian noise.*

*Proof.* The forward and backward models for the mean-variance HNMs are given by

$$X_2 = m_2(B_{\mathrm{pa}^M(2)}^M X_1) + v_2(B_{\mathrm{pa}^V(2)}^V X_1)E_2 = m_2^O(X_1) + v_2^O(X_1)E_2, \tag{14}$$

$$X_1 = m_1(B_{\mathrm{pa}^M(1)}^M X_2) + v_1(B_{\mathrm{pa}^V(1)}^V X_2)E_1 = m_1^O(X_2) + v_1^O(X_2)E_1, \tag{15}$$

where $m_j^O(X_j) := m_j(B_{\mathrm{pa}^M(j)}^M X_{\mathrm{pa}(j)})$ and $v_j^O(x) := v_j(B_{\mathrm{pa}^V(j)}^V X_{\mathrm{pa}(j)})$ ($j = 1, 2$) denote the functions in the original HNM, and $B_{\mathrm{pa}^M(j)}^M, B_{\mathrm{pa}^V(j)}^V \in \{0, 1\}$ are binary masking variables.

Under Condition (c), unidentifiable scenarios are restricted to Scenarios 1 and 2 in Theorem B.1. As with the ANM cases (Peters et al., 2014, Proposition 17), causal minimality (Assumption 3.2) implies the non-constancy of functions and requires that functions $m_j^O$ and $v_j^O$ in Eqs. (14) and (15) not simultaneously be constant with respect to any input variable in $\boldsymbol{X}_{\mathrm{pa}(j)}$. Since this requirement does not allow $B_{\mathrm{pa}^M(j)}^M = B_{\mathrm{pa}^V(j)}^V = 0$, under Conditions (a) and (b) on the mean and variance functions $m_j$ and $v_j$, the functions in the original HNM belong to either of the following three cases:

1. nonlinear $m_j^O$ and constant $v_j^O$ (i.e., $B_{\mathrm{pa}^M(j)}^M = 1$ and $B_{\mathrm{pa}^V(j)}^V = 0$),

2. constant $m_j^O$ and piecewise (**but not constant**) $v_j^O$ (i.e., $B_{\mathrm{pa}^M(j)}^M = 0$ and $B_{\mathrm{pa}^V(j)}^V = 1$),

3. nonlinear $m_j^O$ and piecewise $v_j^O$ (i.e., $B_{\mathrm{pa}^M(j)}^M = B_{\mathrm{pa}^V(j)}^V = 1$).

In Case 1, nonlinear $m_j^O$ and constant $v_j^O$ exclude Scenario 2 and 1, respectively. In Case 2, piecewise but not constant $v_j^O$ suffices to exclude both scenarios. In Case 3, both scenarios are excluded in the same way as Case 1. Hence, all three cases exclude Scenarios 1 and 2. This proves Theorem B.3. $\square$

### B.1.2    MULTIVARIATE CASE ($d > 2$)

We extend the bivariate identifiability result in Theorem B.3 to the multivariate case $d > 2$:

**Theorem B.4:** *Under Assumptions 3.1, 3.2, 3.3 and 3.4, moment-agnostic causal graph $G$ of the mean-variance HNM is identifiable from distribution $\mathrm{P}(\boldsymbol{X})$ if for $j = 1, \ldots, d$, (a) $m_j$ is a nonlinear function, (b) $v_j$ is a piecewise function, and (c) $E_j$ is Gaussian noise.*

The proof proceeds in the same way as that of Yin et al. (2024, Theorem 2), which we describe below.

*Proof.* Suppose the joint distribution $\mathrm{P}(\boldsymbol{X})$ is generated from a mean-variance HNM with moment-agnostic causal graph $G$. Assume also that the same distribution can be obtained from another mean-variance HNM with a different moment-agnostic graph $G'$, i.e., $G \neq G'$. Assumptions 3.3 and 3.4 imply that $G$ and $G'$ are both DAGs (Corollary A.1). Therefore, under Assumptions 3.1, and 3.2, $\mathrm{P}(\boldsymbol{X})$ satisfies the causal Markov condition and causal minimality with respect to $G$ and $G'$. In such cases, according to Proposition 29 of Peters et al. (2014), there is a pair of variables, $L, Y \in \boldsymbol{X}$, such that for three variable subsets, $\boldsymbol{Q} = \boldsymbol{X}_{\mathrm{pa}^G(Y)} \backslash \{L\}$, $\boldsymbol{R} = \boldsymbol{X}_{\mathrm{pa}^{G'}(L)} \backslash \{Y\}$, and $\boldsymbol{S} = \boldsymbol{Q} \cup \boldsymbol{R}$, the following conditions hold:

$$L \to Y \text{ in } G; \quad L \leftarrow Y \text{ in } G' \tag{16}$$

$$\boldsymbol{S} \subseteq \boldsymbol{X}_{\mathrm{nd}^G(Y)} \backslash \{L\}; \quad \boldsymbol{S} \subseteq \boldsymbol{X}_{\mathrm{nd}^{G'}(L)} \backslash \{Y\}, \tag{17}$$

where $\boldsymbol{X}_{\mathrm{pa}^G(Y)}$ and $\boldsymbol{X}_{\mathrm{pa}^{G'}(L)}$ are the variables expressed as the parents of $Y$ in $G$ and $L$ in $G'$, respectively, and $\boldsymbol{X}_{\mathrm{nd}^G(Y)}$ and $\boldsymbol{X}_{\mathrm{nd}^{G'}(L)}$ are the variables represented as the non-descendants of $Y$ in $G$ and of $L$ in $G'$, respectively. From Lemma 37 in Peters et al. (2014), if the structural equation of endogeneous variable $X_j \in \boldsymbol{X}$ ($j \in \{1, \ldots, d\}$) is given by the following general formulation with scalar exogenous variable $E_j$

$$X_j = f_j(\boldsymbol{X}_{\mathrm{pa}^G(j)}, E_j) \quad \text{for } j = 1, \ldots, d, \tag{18}$$

then independence relation $X_j \perp\!\!\!\perp \boldsymbol{K}$ holds for variable subset $\boldsymbol{K} \subseteq \boldsymbol{X}_{\mathrm{nd}^G(j)}$. Since the structural equation of the mean-variance HNM is a special case of (18), we can apply this result to derive the following independence relations:

$$E_Y \perp\!\!\!\perp \{L, \boldsymbol{S}\} \quad \text{and} \quad E_L \perp\!\!\!\perp \{Y, \boldsymbol{S}\}. \tag{19}$$

Consider the conditioning of random variables $\boldsymbol{S}$ on $\boldsymbol{S} = \boldsymbol{s}$ with $\mathrm{P}(\boldsymbol{s}) > 0$. Let $Y^*$ and $L^*$ be random variables obeying conditional distributions $\mathrm{P}(Y \mid \boldsymbol{S} = \boldsymbol{s})$ and $\mathrm{P}(L \mid \boldsymbol{S} = \boldsymbol{s})$, respectively. According to Lemma 36 in Peters et al. (2014), if the independence relations in (19) hold, then the values of $Y^*$ and $L^*$ are determined by

$$Y^* = f_Y(L^*, \boldsymbol{q}, E_Y); \tag{20}$$

$$L^* = f_L(Y^*, \boldsymbol{r}, E_L), \tag{21}$$

where $\boldsymbol{q}, \boldsymbol{r} \in \boldsymbol{s}$ are the values of variables $\boldsymbol{Q}$ and $\boldsymbol{R}$. Eqs. (20) and (21) imply that joint distribution $\mathrm{P}(Y^*, L^*)$ is compatible with both the forward and backward models. However, this contradicts the bivariate identifiability result in Theorem B.3, which rules out such coexistence under the mean-variance HNM. Thus, $G'$ cannot differ from $G$, and $G$ is uniquely identifiable from $\mathrm{P}(\boldsymbol{X})$. □

*Remark B.5.* Conditions (a), (b), and (c) in Theorem B.4 are weaker than conditions (A), (B), and (C) in Theorem 3.5: Variance function $v_j$ can be a constant under the former but cannot be under the latter. Therefore, conditions (A), (B), and (C) in Theorem 3.5 are also sufficient to ensure identifiability, meaning that they are also the identifiability conditions for moment-agnostic causal graph $G$.

## B.2 IDENTIFIABILITY OF MEAN AND VARIANCE CAUSAL GRAPHS OF MEAN-VARIANCE HNM

Based on the identifiability of moment-agnostic causal graph $G$ in Theorem B.4, we prove that the mean and variance causal graphs $G^M$ and $G^V$ are also identifiable under the stronger conditions.

In particular, we need additional conditions where **both** mean and variance functions in the mean-variance HNM are not constant with respect to the inputs. As noted in Remark 3.6, excluding such constant cases is essential to detect the presence of the inputs of the mean and variance functions. Since nonlinearity condition (a) in Theorem B.4 already requires that mean function $m_j$ is not a constant function, we only have to add the non-constant condition on variance function $v_j$:

**Theorem 3.5:** *Under Assumptions 3.1, 3.2, 3.3, and 3.4, mean and variance causal graphs $G^M$ and $G^V$ are identifiable from observational distribution $\mathrm{P}(\boldsymbol{X})$ if for $j = 1, \ldots, d$, (A) $m_j$ is a nonlinear function, (B) $v_j$ is a piecewise function, **but not a constant function**, and (C) $E_j$ is a Gaussian noise.*

*Proof.* From Theorem B.4 (and Remark B.5), Conditions (A), (B), and (C) are sufficient to identify the moment-agnostic causal graph. Hence, it is sufficient to show the identifiability of mask matrices

$\mathbf{B}_j^{\mathcal{I}} \in \{0,1\}^{|\mathcal{I}| \times |\mathrm{pa}(j)|}$ for separating a superset of parental variables $\boldsymbol{X}_{\mathrm{pa}(j)} = \boldsymbol{X}_{\mathrm{pa}^M(j)} \cup \boldsymbol{X}_{\mathrm{pa}^V(j)}$ into mean and variance causes $\boldsymbol{X}_{\mathrm{pa}^M(j)} = \mathbf{B}_j^{\mathrm{pa}^M(j)} \boldsymbol{X}_{\mathrm{pa}(j)}$ and $\boldsymbol{X}_{\mathrm{pa}^V(j)} = \mathbf{B}_j^{\mathrm{pa}^V(j)} \boldsymbol{X}_{\mathrm{pa}(j)}$, for each $j = 1, \ldots, d$.

As discussed in Section 3.1, each endogenous variable $X_j$ in the mean-variance HNM follows a conditional Gaussian:

$$X_j \mid \boldsymbol{X}_{\mathrm{pa}(j)} \sim \mathcal{N}(m_j(\boldsymbol{X}_{\mathrm{pa}^M(j)}), c_{E_j} \cdot (v_j(\boldsymbol{X}_{\mathrm{pa}^V(j)}))^2), \tag{22}$$

where $c_{E_j} := \mathbb{V}[E_j]$ is the variance of zero-mean noise $E_j$. Using mask matrices, the conditional Gaussian in (22) can be equivalently rewritten as

$$X_j \mid \boldsymbol{X}_{\mathrm{pa}(j)} \sim \mathcal{N}(m_j(\mathbf{B}_j^{\mathrm{pa}^M(j)} \boldsymbol{X}_{\mathrm{pa}(j)}), c_{E_j} \cdot (v_j(\mathbf{B}_j^{\mathrm{pa}^V(j)} \boldsymbol{X}_{\mathrm{pa}(j)}))^2). \tag{23}$$

Suppose that under different mean and variance functions, this conditional Gaussian distribution can be identical for different parental variables $\boldsymbol{X}_{\mathrm{pa}^M(j)} \neq \boldsymbol{X}_{\mathrm{pa}'^M(j)}$ and $\boldsymbol{X}_{\mathrm{pa}^V(j)} \neq \boldsymbol{X}_{\mathrm{pa}'^V(j)}$:

$$\begin{aligned}
&\mathcal{N}(m_j(\mathbf{B}_j^{\boldsymbol{X}_{\mathrm{pa}^M(j)}} \boldsymbol{X}_{\mathrm{pa}(j)}), c_{E_j} \cdot (v_j(\mathbf{B}_j^{\boldsymbol{X}_{\mathrm{pa}^V(j)}} \boldsymbol{X}_{\mathrm{pa}(j)}))^2) \\
&= \mathcal{N}(m_j'(\mathbf{B}_j^{\boldsymbol{X}_{\mathrm{pa}'^M(j)}} \boldsymbol{X}_{\mathrm{pa}(j)}), c_{E_j} \cdot (v_j'(\mathbf{B}_j^{\boldsymbol{X}_{\mathrm{pa}'^V(j)}} \boldsymbol{X}_{\mathrm{pa}(j)}))^2),
\end{aligned} \tag{24}$$

where $m_j \neq m_j'$, $v_j \neq v_j'$, $\mathbf{B}_j^{\boldsymbol{X}_{\mathrm{pa}^M(j)}} \neq \mathbf{B}_j^{\boldsymbol{X}_{\mathrm{pa}'^M(j)}}$, and $\mathbf{B}_j^{\boldsymbol{X}_{\mathrm{pa}^V(j)}} \neq \mathbf{B}_j^{\boldsymbol{X}_{\mathrm{pa}'^V(j)}}$. Distributional equality (24) implies that the conditional mean and variance are identical. Formally, for any $\boldsymbol{X}_{\mathrm{pa}(j)}$'s values and for some $m_j \neq m_j'$, $v_j \neq v_j'$, $\mathbf{B}_j^{\boldsymbol{X}_{\mathrm{pa}^M(j)}} \neq \mathbf{B}_j^{\boldsymbol{X}_{\mathrm{pa}'^M(j)}}$, and $\mathbf{B}_j^{\boldsymbol{X}_{\mathrm{pa}^V(j)}} \neq \mathbf{B}_j^{\boldsymbol{X}_{\mathrm{pa}'^V(j)}}$, the following equalities must hold:

$$m_j(\mathbf{B}_j^{\boldsymbol{X}_{\mathrm{pa}^M(j)}} \boldsymbol{X}_{\mathrm{pa}(j)}) = m_j'(\mathbf{B}_j^{\boldsymbol{X}_{\mathrm{pa}'^M(j)}} \boldsymbol{X}_{\mathrm{pa}(j)}) \tag{25}$$

$$c_{E_j} \cdot (v_j(\mathbf{B}_j^{\boldsymbol{X}_{\mathrm{pa}^V(j)}} \boldsymbol{X}_{\mathrm{pa}(j)}))^2 = c_{E_j} \cdot (v_j'(\mathbf{B}_j^{\boldsymbol{X}_{\mathrm{pa}'^V(j)}} \boldsymbol{X}_{\mathrm{pa}(j)}))^2. \tag{26}$$

Since $\mathbf{B}_j^{\boldsymbol{X}_{\mathrm{pa}^M(j)}} \neq \mathbf{B}_j^{\boldsymbol{X}_{\mathrm{pa}'^M(j)}}$, there exists at least one variable that is included in one of $\boldsymbol{X}_{\mathrm{pa}^M(j)} = \mathbf{B}_j^{\mathrm{pa}^M(j)} \boldsymbol{X}_{\mathrm{pa}(j)}$ and $\boldsymbol{X}_{\mathrm{pa}'^M(j)} = \mathbf{B}_j^{\boldsymbol{X}_{\mathrm{pa}'^M(j)}} \boldsymbol{X}_{\mathrm{pa}(j)}$, but not in the other. Assume, without loss of generality, that $\boldsymbol{X}_{\mathrm{pa}'^M(j)}$ contains the variables that are missing in $\boldsymbol{X}_{\mathrm{pa}^M(j)}$. Those variables are not inputs of function $m_j$, yet Eq. (25) must hold for any $\boldsymbol{X}_{\mathrm{pa}(j)}$'s values. Therefore, these additional variables must have **no influence on the output of function** $m_j'$, implying that $m_j'$ must be constant with respect to these variables. However, this violates the non-constant conditions on $m_j'$. A similar discussion holds for variance function $v_j$ in (26). Thus we prove the identifiability of mean and variance causal graphs $G^M$ and $G^V$. $\qquad \square$

## C  OVERVIEW OF DIFFERENTIABLE SAMPLING MODELS

### C.1  GUMBEL SOFTMAX DISTRIBUTION

The Gumbel Softmax distribution (Jang et al., 2017) provides a differentiable approximation for addressing the non-differentiability of the sampling operation of categorical variables. This approximation is needed to learn the parameters of neural-network-based discrete sampling models by propagating gradients in standard backpropagation.

The core idea lies in the traditional statistical technique called *Gumbel-Max trick* (Gumbel, 1954), which performs sampling from a $C$-class categorical distribution with log probabilities $[\phi_0, \ldots, \phi_{C-1}]$ ($C \geq 2$) by taking argmax over the log probabilities perturbed by Gumbel noise $g_c \sim \mathrm{Gumbel}(0)$:

$$Z = \underset{c \in \{0, \ldots, C-1\}}{\arg\max} \{\phi_c + g_c\}, \tag{27}$$

where $c \in \{0, \ldots, C-1\}$ denotes a class index.

The argmax operator of Gumbel-Max trick in (27) is non-differentiable with respect to parameters $\phi_0, \ldots, \phi_{C-1}$, thus hindering gradient-based parameter learning. To make this argmax operator

differentiable, Gumbel-Softmax trick replaces it with a differentiable softmax function:

$$\tilde{Z}_c = \frac{e^{(\log \phi_c + g_c)/\tau}}{\sum_{c'=0}^{C-1} e^{(\log \phi_{c'} + g_{c'})/\tau}} \in [0,1], \tag{28}$$

where $g_c \sim \text{Gumbel}(0)$ is a standard Gumbel noise, and $\tau$ is a temperature parameter that controls the smoothness of the softmax function. As $\tau \to 0$, vector $[\tilde{Z}_0, \ldots, \tilde{Z}_{C-1}]$ approaches a one-hot vector, mimicking a categorical sample.

In this paper, we employ the Gumbel-Softmax distribution model with number of classes $C = 2$ as a sampling model for upper-triangular matrix elements $U_{i,j}^M, U_{i,j}^V \in \{0,1\}$ for $i, j \in \{1, \ldots, d\}$.

### C.2 PERMUTATION SAMPLING WITH SOFTSORT FUNCTION

Charpentier et al. (2022) sample the continuous relaxation of a $d \times d$ permutation matrix by dealing with the non-differentiability of the categorical sampling operation over $\{1, \ldots, d\}$.

Their sampling procedure is founded on the Gumbel Top-$K$ trick (Vieira, 2014), which samples $K$ categorical values over $\{1, \ldots, d\}$ ($d \geq 2$) without replacement by iteratively selecting the top $K$ values of the logits perturbed by standard Gumbel noise:

$$I_1, \ldots, I_K = \underset{c \in \{1, \ldots, d\}}{\arg \text{top-K}} \{\log \psi_c + g_c\}, \tag{29}$$

where $I_1, \ldots, I_K \in \{1, \ldots, d\}$ are the sampled class indices, arg top-K denotes an operation that selects the $K$ largest values from the inputs, $\psi_c$ is the probability of the categorical distribution for class $c \in \{1, \ldots, d\}$, and $g_c \sim \text{Gumbel}(0)$ is the standard Gumbel noise. By setting $K = d$, Eq. (29) can be used to obtain a permutation that sorts vector $[\log \psi_1 + g_1, \ldots, \log \psi_d + g_d]$ in ascending order. However, the arg top-K operator is not differentiable, as with the argmax operator in (27).

For this reason, Charpentier et al. (2022) replace the arg top-K in (29) with a SoftSort function (Prillo and Eisenschlos, 2020), which is differentiable with respect to its inputs. This function is defined by applying the softmax function to a pairwise distance matrix as

$$\text{SoftSort}_\tau(\boldsymbol{v}) = \text{softmax}\left(-\frac{d(\boldsymbol{v}, \text{sort}(\boldsymbol{v}))}{\tau}\right), \tag{30}$$

where $d(\boldsymbol{v}, \boldsymbol{v}')$ is a differentiable semi-metric function (e.g., the L1 distance $d(x, y) = \|\boldsymbol{v} - \boldsymbol{v}'\|_1$) that returns a pairwise distance matrix, $\text{sort}(\boldsymbol{v})$ denotes an operation that rearranges the elements of vector $\boldsymbol{v}$ in ascending order, and $\tau$ is a hyperparameter that controls the smoothness. By applying a SoftSort function in (30) to the perturbed log probabilities, Charpentier et al. (2022) sample a continuous relaxation of permutation matrix, i.e., $\tilde{\boldsymbol{\Pi}} \in \mathbb{R}^{d \times d}$.

As described in Section 4.1.1, we directly employ the above sampling procedure for permutation matrix $\boldsymbol{\Pi}$.

## D EXPERIMENTAL SETTINGS

### D.1 BASELINES

In our experiments, we compare the performance of our method with the following baselines:

- Metropolice-coupled Markov chain Monte Carlo (**MC3**) (Madigan et al., 1995; Giudici and Castelo, 2003), which performs a Metropolis-Hastings-based sampling whose target distribution is the posterior distribution over Bayesian network structures. We consider a tractable posterior formulation with the linear Gaussian models and employ the R-based implementation downloaded from `https://github.com/rjbgoudie/structmcmc` (Goudie and Mukherjee, 2016), which is published under GNU General Public License v3 (GPL-3).

- Differentiable DAG sampling (**DDS**) (Charpentier et al., 2022), which approximately infers the posterior over a single causal DAG by solving a variational inference problem. Based on non-linear ANM formulation, it evaluates the expected score of each DAG using the squared reconstruction loss.

- Identifiable causal discovery under heteroscedastic data (**ICDH**) (Yin et al., 2024), which learns the weighted adjacency matrix of a moment-agnostic causal graph based on a nonlinear HNM to output a single point estimate. We employ the original source codes on `https://github.com/naiyuyin/ICDH` and set the threshold for the inferred weighted adjacency matrix to 0.2.

- Heteroscedastic causal structure learning (**HOST**) (Duong and Nguyen, 2023), which offers a point estimate of a causal graph by performing conditional independence testing based on a nonlinear HNM. We use the implementation downloaded from `https://github.com/baosws/HOST` and set the significance level of the conditional independence test to $\alpha = 0.05$.

### D.2   Settings of Each Method

- **SCM parameterization:** For our method, we parameterize the mean and variance functions $m_j$ and $v_j$ of our mean-variance HNM in (3) using 2-layered MLPs. The nonlinear functions for **DDS**, **ICDH**, and **HOST** are parameterized using the MLPs by following their default settings.

- **Hyperparameter tuning:** For each method, we tune the hyperparameters by conducting a grid search based on objective function values. Regarding our method, we conduct a search over batch size $B \in \{32, 64, 128, 256\}$ and the number of hidden neurons $h \in \{8, 16, 32, 64\}$ in the 2-layered MLPs. To do so, we use the `optuna` package (Akiba et al., 2019) in Python Package Index (PyPI) (licensed under GNU GPL-2) with the fixed random seed.

- **Optimization algorithm:** For our method, we can use any gradient-based optimization algorithm. In our experiments, following the ICDH method (Yin et al., 2024), we use the Limited-memory Broyden–Fletcher–Goldfarb–Shanno (LBFGS) algorithm, which is a popular optimization scheme in the family of quasi-Newton methods. We employ the LBFGS wrapper for Pytorch on `https://gist.github.com/arthurmensch/c55ac413868550f89225a0b9212aa4cd`, which is released under the MIT License and provided on an "AS IS" basis. To implement our prior knowledge incorporation approach presented in Section 4.3, we employ the `quadprog` package in Python Package Index (PyPI) (licensed under GNU GPL-2).

### D.3   Synthetic and Semi-Synthetic Data Generation Processes

This section describes the generation processes of the synthetic and semi-synthetic data used in Section 5.1.

#### D.3.1   Random Causal Graph Generation

We randomly sample the ground truth mean and variance causal graphs $G^M$ and $G^V$.

In synthetic data generation, we randomly sample the ground truth mean and variance causal graphs using an ER model with 1 expected edge per node for $d = 5$ and 2 expected edges per node for $d = 10, 20$, and $50$.[2]

In semi-synthetic data generation, since real-world gene regulatory networks often exhibit scale-free properties (Albert, 2005), we randomly draw the ground truth mean and variance causal graphs using the SF model. For all $d = 5, 10, 20, 50$, we set their expected degree to 2.

To guarantee the acyclicity of the sampled adjacency matrices, we obtain upper-triangular matrices $\mathbf{U}^M$ and $\mathbf{U}^V$ by masking the lower-triangular elements of the sampled adjacency matrices and randomly permuting the node indices.

#### D.3.2   Data Generation

Using the random causal graphs described in Appendix D.3.1, we generate the synthetic and semi-synthetic datasets.

Regarding the synthetic datasets, we follow Yin et al. (2024) and sample the data from a mean-variance HNM parameterized with randomly initialized MLPs. To sample the values of variable $X_j$

---

[2]For $d = 5$, we use an ER model with 1 expected edge, not 2 expected edges, because the ER model with 2 expected edges always produces a complete graph for $d = 5$, thus losing randomness.

$(j \in \{1, \ldots, d\})$, we formulate the structural equation as

$$X_j = m_j \left( \boldsymbol{X}_{\mathrm{pa}^M(j)} \right) + \mathrm{e}^{v_j \left( \boldsymbol{X}_{\mathrm{pa}^V(j)} \right)} E_j, \tag{31}$$

where $m_j(\cdot)$ and $v_j(\cdot)$ are MLPs with randomly initialized parameters, and $E_j \sim \mathcal{N}(0, 1)$ is noise that follows the standard Gaussian distribution. Following Yin et al. (2024), we parameterize MLPs $m_j(\cdot)$ and $v_j(\cdot)$ using 1 hidden layer with 100 neurons and sigmoid activation.

To generate semi-synthetic datasets, we download the SERGIO simulator from `https://github.com/PayamDiba/SERGIO` (Dibaeinia and Sinha, 2020) (licensed under GNU GPL-3) and modify it such that the ground truth mean and variance causal graphs are different. As described in Section 5.1, this simulator simulates the gene expression data by sampling from the steady state of a stochastic differential equation (called a chemical Langevin equation (CLE)) that represents the rate of the biochemical reactions in a gene regulatory network. Since this differential equation consists of both mean and additive noise terms based on a zero-mean white noise Gaussian processes, and both are affected by the identical set of parental variables (see the formulation detail in (Dibaeinia and Sinha, 2020)), it can be regarded as a special case of the original HNM. For this reason, to obtain the datasets based on the mean-variance HNM, we use different parental variables between the mean and noise terms in this differential equation and perform sampling from its steady state.

## D.4 REAL-WORLD DATA

We download the Sachs dataset from the link `https://ln5.sync.com/dl/b442986b0#5xpiy2n2-q9j87qze-kydrb7wn-xgqjiw2c`, which is shared by Charpentier et al. (2022).

## D.5 PERFORMANCE METRICS

We evaluate the performance of each method with the following widely used metrics:

- Structural Hamming distance (**SHD**), which measures the number of edge additions, removals, and reversals to turn the inferred causal graph into a ground truth causal graph. In Figure 2, we report the **SHD error rate**, defined as each method's SHD divided by the maximum possible SHD, i.e., $\mathrm{SHD}/\binom{d}{2}$, where $d$ is the number of nodes.

- Expected structural Hamming distance ($\mathbb{E}$-**SHD**), which is an expected value of SHD. Let $\hat{\mathrm{P}}(\mathbf{A} \mid \mathcal{D})$ be an inferred posterior over DAG adjacency matrix $\mathbf{A}$. Then the expected SHD is estimated using $n_{\mathrm{MC}}$ Monte Carlo samples as

$$\mathbb{E}\text{-}\mathbf{SHD} \simeq \frac{1}{n_{\mathrm{MC}}} \sum_{i=1}^{n_{\mathrm{MC}}} \mathbf{SHD}(\mathbf{A}^{(i)}, \mathbf{A}^{\mathrm{True}}), \tag{32}$$

where $\mathbf{A}^{\mathrm{True}}$ is the ground truth causal graph, and $\mathbf{A}^{(1)}, \ldots, \mathbf{A}^{(n_{\mathrm{MC}})} \sim \hat{\mathrm{P}}(\mathbf{A} \mid \mathcal{D})$ are the Monte Carlo samples drawn from inferred posterior $\hat{\mathrm{P}}(\mathbf{A} \mid \mathcal{D})$. We set the number of Monte Carlo samples to $n_{\mathrm{MC}} = 2000$.

- F1 score, which compares the presence of each inferred edge with the ground truth edge set.

As described in Section 5, we evaluate $\mathbb{E}$-**SHD** for the Bayesian methods (**Proposed**, **DDS**, and **DiBS**) and compute **SHD** for the point estimation methods (**ICDH** and **HOST**).

## D.6 COMPUTING INFRASTRUCTURE

For our experiments, we used a 64-bit Ubuntu machine with 2.9GHz AMD EPYC 7513 32-core (x2) CPUs, NVIDIA RTX A6000 PCI-EX16 Gen4 (x8) GPUs, and 512-GB RAM.

# E ADDITIONAL EXPERIMENTAL RESULTS

## E.1 PERFORMANCE OF INFERRING DENSE MEAN AND VARIANCE CAUSAL GRAPHS

This section presents the performance in cases where the mean and variance causal graphs are dense.

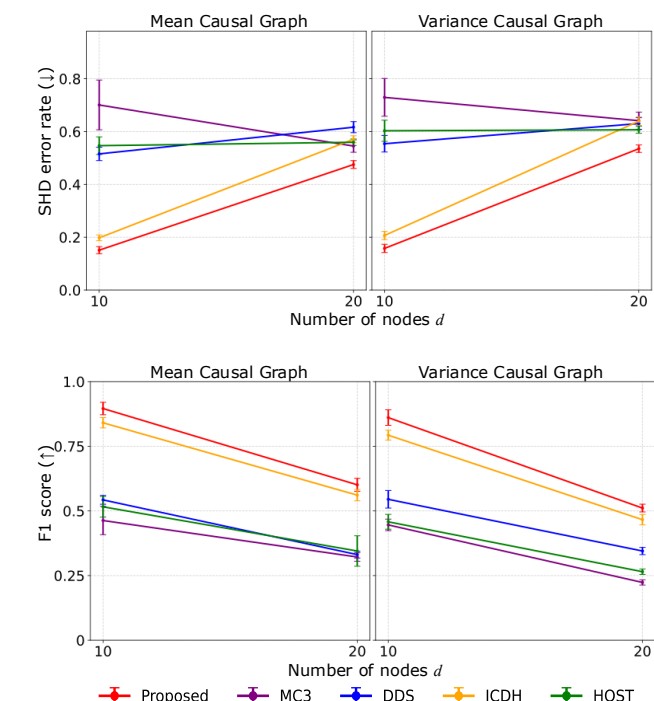

Figure 3: Mean and variance causal graph inference performance under dense graph settings (ER model with 4 expected edges per node). Achieving **both** lower SHD rate (top) and higher F1 score (bottom) is better.

### E.1.1 DATA

We create synthetic datasets in the same way as Appendix D.3, except that the random causal graphs are generated from the ER model with 4 expected edges per node for $d = 10, 20$.

### E.1.2 RESULTS

Figure 3 presents the results. Once again, our method outperforms the baselines in terms of both SHD rate and F1 score. Compared with the synthetic data experiments presented in Section 5.1, inferring dense mean and variance causal graphs is more challenging, as indicated by the higher SHD values for all methods when $d = 20$. However, our method achieves the best F1 score while maintaining lowest SHD values, demonstrating the effectiveness of our method for the moment-driven causal graph discovery task even under dense mean and variance causal graphs.

### E.2 MOMENT-AGNOSTIC CAUSAL GRAPH INFERENCE PERFORMANCE ON ANM AND HNM DATASETS

### E.2.1 DATA

We create ANM and HNM datasets by randomly generating a ground truth moment-agnostic causal graph from the ER model, where the expected number of edges is given in the same way as Appendix D.3.1.

We generate the ANM datasets by considering the structural equation for each variable $X_j$ ($j \in \{1, \ldots, d\}$):

$$X_j = m_j(\boldsymbol{X}_{\mathrm{pa}(j)}) + E_j, \tag{33}$$

where $m_j(\cdot)$ is a nonlinear function parameterized with a randomly initialized MLP, and $E_j \sim \mathcal{N}(0, \sigma_j^2)$ is a zero-mean Gaussian noise whose variance is sampled from uniform distribution by $\sigma_j^2 \sim \mathrm{U}(0.5, 2)$.

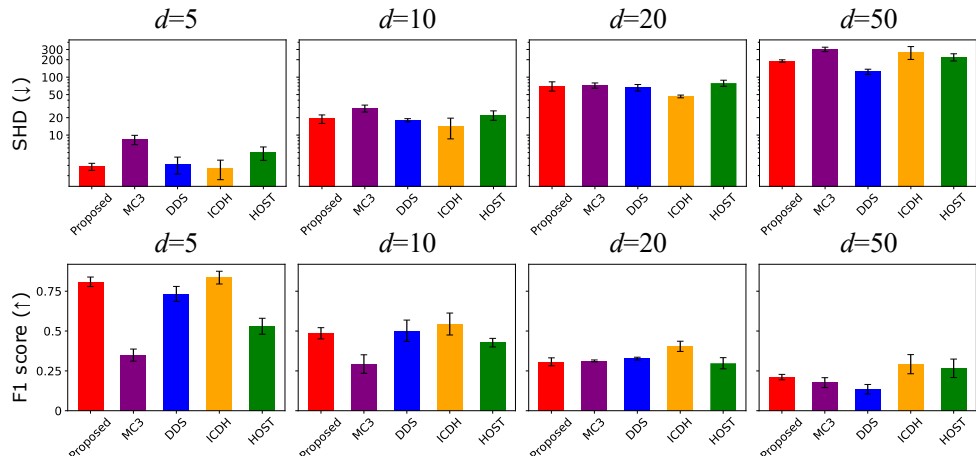

Figure 4: Moment-agnostic causal graph inference performance on ANM datasets with number of nodes $d = 5, 10, 20, 50$: ↓ and ↑ denote "lower is better" and "higher is better".

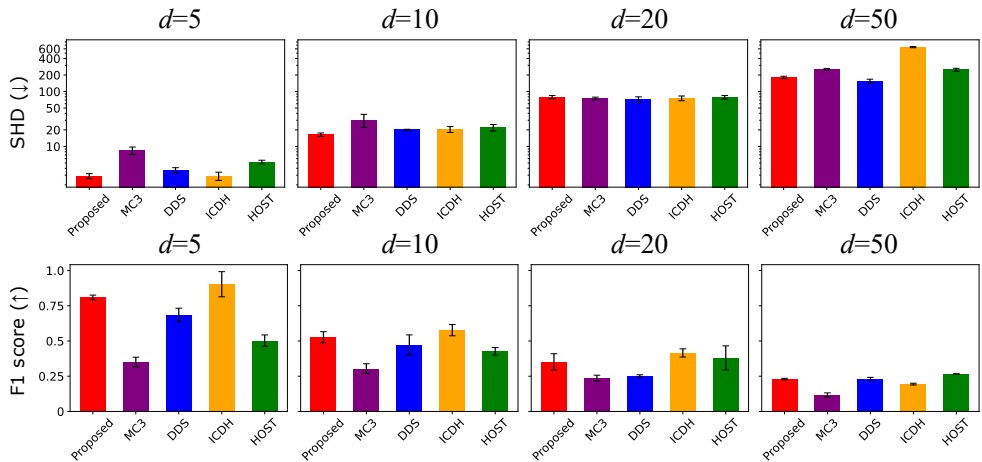

Figure 5: Moment-agnostic causal graph inference performance on HNM datasets with number of nodes $d = 5, 10, 20, 50$: ↓ and ↑ denote "lower is better" and "higher is better".

Regarding the HNM datasets, we parameterize the HNM structural equation:

$$X_j = m_j\left(\boldsymbol{X}_{\mathrm{pa}(j)}\right) + \mathrm{e}^{v_j\left(\boldsymbol{X}_{\mathrm{pa}(j)}\right)} E_j, \tag{34}$$

using randomly initialized MLPs as $m_j(\cdot)$ and $v_j(\cdot)$ and standard Gaussian noise $E_j \sim \mathcal{N}(0, 1)$. We formulate each MLP in the same way as in Appendix D.3.2.

### E.2.2    RESULTS

Figures 4 and 5 present the performance on the ANM and HNM datasets, respectively. Although our method is not designed for moment-agnostic causal graph inference, it achieves comparable performance to the baselines tailored for the ANMs and the HNMs. These results demonstrate that despite the model complexity for inferring the mean and variance causal graphs, our method effectively learns the model parameters by leveraging the differentiable model formulation (Section 4.1.1) and the heteroscedastic noise regression techniques (Section 4.2), thus successfully inferring the moment-agnostic causal graph structure.

Table 2: Estimated posterior probability of MEK $\rightarrow$ ERK in inferred variance causal graph structure

| SAMPLE SIZE | $n = 100$ | $n = 200$ | $n = 853$ |
|---|---|---|---|
| PROPOSED | $0.585 \pm 0.009$ | $0.592 \pm 0.008$ | $0.620 \pm 0.019$ |

### E.3 REAL-WORLD CASE STUDY

This section presents the real-world case study for demonstrating the practical applicability of our moment-driven causal discovery framework. In particular, we evaluate the performance of our method in inferring the variance causal graph structure using the real-world dataset.

#### E.3.1 DATA

We use the Sachs dataset (also employed in Section 5.2) to investigate the biological origin of heteroscedasticity in protein signaling network, which is well documented in the literature (Jacob et al., 2002; Filippi et al., 2016; Davies et al., 2020).

Although the complete ground-truth for both mean and variance causal structures is unknown in the Sachs dataset, prior literature (Filippi et al., 2016) has identified mitogen-activated protein kinase (MEK) as a driver of variance in extracellular signal-regulated kinase (ERK) expression. That is, the directed edge MEK $\rightarrow$ ERK is expected to appear in the ground-truth variance causal graph.

Before applying our method, we empirically validate the statistical significance of heteroscedasticity (i.e., a variance-level association, informative for but not by itself establishing causality) between MEK and ERK in the Sachs dataset. To do so, we perform a heteroscedasticity test called the White test, using the residuals of a random forest regression. The results show that ERK variance varies significantly with MEK expression levels with $p$-value $< 0.0096$, thus confirming that the heteroscedasticity between MEK and ERK is statistically significant.

#### E.3.2 RESULTS

We apply our method to the Sachs dataset and compute the posterior probability of the edge MEK $\rightarrow$ ERK in the inferred variance causal graph.

Table 2 presents the mean and standard deviation of the estimated posterior probability across 20 runs. The estimated posterior probability is consistently high, even in low-sample regimes. Although accurately inferring the complete graph structure from such small-sample data is challenging as shown in Table 1 in Section 5.2, our Bayesian inference framework effectively captures the significant heteroscedasticity between MEK and ERK and successfully identifies the directed edge MEK $\rightarrow$ ERK in the inferred variance causal graph structure, as shown in biological findings (Filippi et al., 2016).

Importantly, these results demonstrate the practical applicability of our Bayesian causal discovery framework for moment-driven causal discovery, which is designed to model inference uncertainty and to remain effective even with small data.

### E.4 RUN TIME COMPARISON

We compare the run time of our method using the Sachs dataset with sample size $n = 853$. To ensure a fair comparison that excludes implementation-level differences, we choose DDS as a baseline because it also employs a differentiable DAG sampling model.

Table 3 reports the run time of both methods. Our method takes approximately twice as long as DDS, primarily due to the training of additional parameters for the variance causal graph and the variance functions. When incorporating prior knowledge about pairwise node orderings, our method incurs additional computation to run a QP solver for solving the constrained least squares in Eq. (11). This overhead, which could be further reduced by employing more efficient QP solvers, is sufficiently small to be acceptable in practice, given that it enables high causal graph inference performance on limited data (as shown in the results when $n = 100$ and 200 in Table 1 in Section 5.2).

Table 3: Run time comparison on Sachs dataset ($n = 853$)

|  | RUN TIME [SEC] |
| --- | --- |
| DDS | $367 \pm 11$ |
| PROPOSED | $733 \pm 19$ |
| PROPOSED $+25\%$ | $852 \pm 38$ |
| PROPOSED $+50\%$ | $1260 \pm 51$ |

Table 4: SHD scores on synthetic datasets with number of nodes $d = 5, 10$ and sample size $n = 500$.

| METHOD | MEAN ($d = 5$) | VARIANCE ($d = 5$) | MEAN ($d = 10$) | VARINACE ($d = 10$) |
| --- | --- | --- | --- | --- |
| PROPOSED | $\mathbf{2.34 \pm 0.20}$ | $\mathbf{3.33 \pm 0.30}$ | $\mathbf{15.5 \pm 1.2}$ | $\mathbf{19.3 \pm 1.1}$ |
| ICDH | $5.05 \pm 0.42$ | $4.55 \pm 0.40$ | $20.1 \pm 1.1$ | $20.9 \pm 1.2$ |
| HOST | $4.84 \pm 0.49$ | $4.60 \pm 0.45$ | $19.6 \pm 1.8$ | $21.6 \pm 1.4$ |
| VARSORT | $5.00 \pm 1.00$ | $6.00 \pm 1.66$ | $18.5 \pm 6.2$ | $21.9 \pm 4.4$ |

### E.5 COMPARISON WITH VARSORT

VarSort method (Reisach et al., 2021), which simply determines the permutation of the causal DAG based on the marginal variance order, is a standard baseline for testing the methods based on ANMs.

However, we exclude Varsort from our main comparison in Section 5.1 and Section 5.2. The reason is identical to the one discussed in Yin et al. (2024): The marginal variance order estimation is unstable due to heteroscedastic noise, leading to large variance in estimated causal node orderings.

Below, we elaborate on this point from a theoretical and empirical perspective.

#### E.5.1 THEORETICAL COMPARISON

We illustrate the formulation difference of marginal variance between the ANM and our mean-variance HNM. From the law of total variance, the marginal variance of a variable $X_j$ following the mean-variance HNM is given by

$$\mathbb{V}[X_j] = \mathbb{V}[\mathbb{E}[X_j \mid \boldsymbol{X}_{\mathrm{pa}^M(j)}]] + \mathbb{E}[\mathbb{V}[X_j \mid \boldsymbol{X}_{\mathrm{pa}^V(j)}]]$$
$$= \mathbb{V}[m_j(\boldsymbol{X}_{\mathrm{pa}^M(j)})] + \mathbb{E}[\big(v_j(\boldsymbol{X}_{\mathrm{pa}^V(j)})\big)^2],$$

where $m_j(\cdot)$ and $v_j(\cdot)$ are the mean and variance functions of the mean-variance HNM, respectively.

Hence, the ANM and mean-variance HNM datasets have the following marginal variance:

- Under homoscedasticity in the ANM datasets, the second term is constant, as the variance function $v_j(\cdot)$ is constant in the ANM (Section 2).
- Under heteroscedasticity in the mean-variance HNM datasets, however, the second term is non-constant, and the order of marginal variances can be substantially fluctuated.

For this reason, we have the following difference. Under the ANM datasets, the estimated marginal variance order can often be aligned with the causal node ordering, which is why the VarSort method serves as an important baseline for such data. Under the mean-variance HNM datasets, however, the estimated marginal variance order is less likely to be aligned with the causal node ordering.

#### E.5.2 EMPIRICAL COMPARISON

We present the empirical comparison results between our method and VarSort, using synthetic, mean-variance HNM datasets (employed in Section 5.1).

Table 4 presents the (expected) SHD over 20 runs. As expected, the performance of VarSort is unstable, with a large estimation variance. These results suggest that its behavior is not robust but may align with ground truth purely by chance, leading to similar average SHD scores.

By contrast, the HNM-based methods, including our method, consistently achieve smaller variances in SHD, demonstrating their robustness in causal graph inference under heteroscedastic noise.

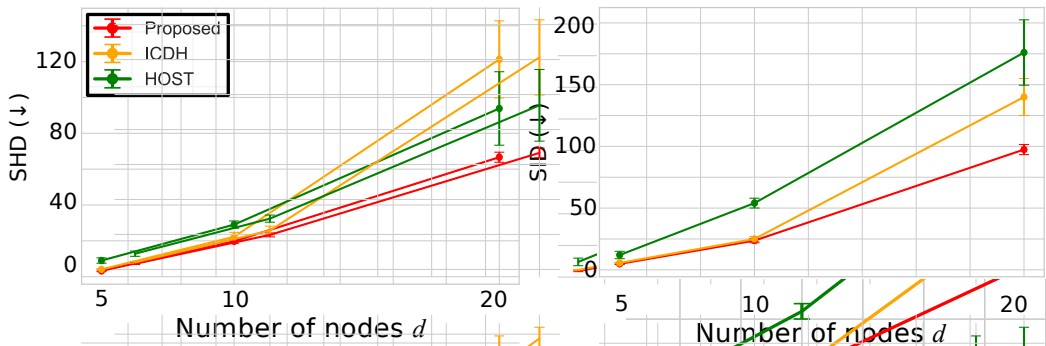

Figure 6: Moment-agnostic causal graph inference performance on synthetic datasets with Laplace noise. ↓ denotes "lower is better".

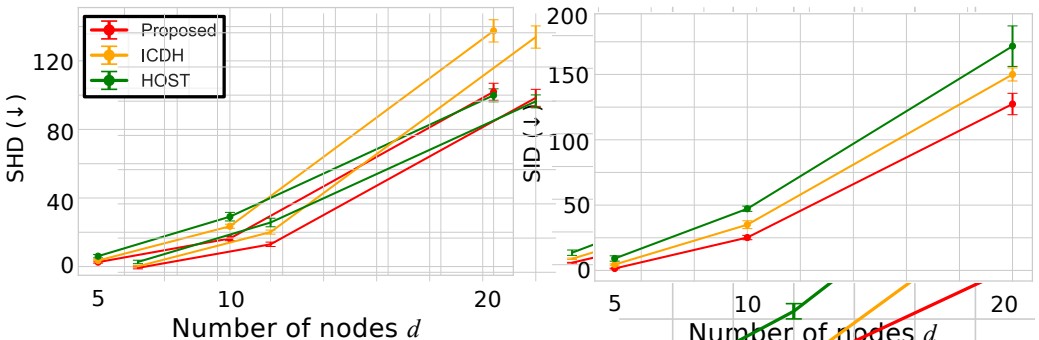

Figure 7: Moment-agnostic causal graph inference performance on synthetic datasets with student-t noise. ↓ denotes "lower is better".

### E.6   PERFORMANCE COMPARISON ON NON-GAUSSIAN SYNTHETIC DATA

We evaluate moment-agnostic causal graph inference performance on non-Gaussian datasets.

#### E.6.1   DATA

We generate synthetic datasets with non-Gaussian heteroscedastic noise by modifying the generation process for the (original) HNM datasets presented in Appendix E.2. In particular, we replace the standard Gaussian noise $E_j \sim \mathcal{N}(0, 1)$ with the following non-Gaussian noises:

- Laplace noise with location 0 and scale 1: $E_j \sim \text{Laplace}(0, 1)$
- Student-t noise with degree of freedom 3: $E_j \sim \text{Student-}t(3)$

#### E.6.2   BASELINES

We compare the performance of our method with **ICDH** and **HOST**, both of which are designed for Gaussian HNMs. This is because most existing methods for non-Gaussian HNMs only address bivariate setup (Immer et al., 2023; Tran et al., 2024). An exception is the SkewScore method by Lin et al. (2025), whose official implementation, however, is not publicly available.

#### E.6.3   METRICS

Along with the (expected) SHD, we use the structural intervention distance (**SID**) (Peters and Bühlmann, 2015), which evaluates how correctly the inferred graph offers adjustment sets, whose marginalization is needed to estimate interventional distributions. Note that from this definition, SID can be applied to measure the quality of the inferred moment-againstic causal graphs, not mean and variance causal graphs.

### E.6.4 RESULTS

Figures 6 and 7 present the results on the datasets with Laplace and student-t noise, respectively.

Despite the likelihood model misspecification, our method consistently outperforms the baselines in terms of both SHD and SID in almost all configurations. This suggests strong empirical robustness to noise misspecification.

A possible reason for this robustness is that our method may mitigate the impact of noise misspecification by downweighting the gradients from extreme observations, owing to our parameter learning strategy including the use of gradient scaling technique (Section 4.2).

