# OpenReview forum: "Moment Matters: Mean and Variance Causal Graph Discovery from Heteroscedastic Data"
_ICLR.cc/2026/Conference — Submitted to ICLR 2026_

### Official Review · Reviewer_LbVq · 2025-10-17

**Soundness:** 3
**Presentation:** 3
**Contribution:** 2
**Rating:** 6
**Confidence:** 3

**Summary:**

This paper proposes a new causal discovery framework that distinguishes between causes affecting the (conditional) mean and those affecting the (conditional) variance of a variable. It introduces the concept of mean and variance causal graphs, extending existing heteroscedastic noise models (HNMs). The authors prove identifiability conditions for these graphs and develop a Bayesian variational inference method that infers their posterior distributions from observational data. Empirical results on synthetic, semi-synthetic, and real datasets show competitive performance compared to prior methods.

**Strengths:**

1. The proposed Bayesian inference with uncertainty quantification is very important, not only for this specific case of discovery in heteroscedastic noise models, but generally to all causal discovery methods. Unlike point-estimation methods, such full posterior estimation gives more robust results especially in small data regimes.

2. Theoretical results and the whole paper progression is generally well structured. The assumptions are stated clearly and used transparently in the derivation of identifiability conditions.

**Weaknesses:**

1. The motivation for separating the mean graph and the variance graph is still unclear to me. While the authors argue for the importance of distinguishing between causes of mean and variance (e.g., in drug design or economic variability), the practical necessity of _explicitly modeling two separate causal graphs_ remains somewhat unclear. Intuitively, one could always first estimate a moment-agnostic causal graph and then use flexible (e.g., nonparametric) regression to analyze how each parent affects the target variable (referred to as "double dipping" in this paper). The necessity could be made more precise and convincing.

2. More literature review and comparison to other HNM models and methods are needed, especially since this is a relatively new area. The authors show two main advantages of their model comparing to others (separating, and Bayesian estimation). But for a balanced comparison, what are the other aspects that other methods may be able to address but not this one? In particular,

   - Are there existing models (e.g., beyond HNM) that allow more general functional forms, such as multiplicative or non-additive noise structures (instead of multiplier only posed on the exogenous noise)?

   - Do any existing methods relax the Gaussian noise assumption?


3. More elaboration about the identifiability conditions are needed. Specifically,

   - Are the stated conditions only sufficient, or are they also necessary for identifiability?

   - What happens when the conditions are violated? For instance, could the authors give simple concrete examples where identifiability fails (e.g., two models with different graphs but same distribution), to illustrate the role of nonlinearity or piecewise variance functions?

4. Minors:

   - The title “Moment Matters” may be slightly misleading to readers expecting techniques involving higher-order moments (e.g., skewness, kurtosis), when in fact the method focuses on first and second moments only (mean and variance). May consider rephrasing or clarifying this early in the introduction.

   - Assumption 3.2 (causal minimality) can be explained more in details in the main body, especially to emphasize that it is weaker than the standard faithfulness assumption.

**Questions:**

see "weaknesses".

---

> ### Author Response · Authors · 2025-11-21
> **Response to Reviewer LbVq**
>
> We cordially thank the reviewer for their time and effort in evaluating our work.
> However, it appears that **their review has been conducted based on an earlier version of our manuscript that was submitted to a different venue in the past**, rather than the current version. As factual evidence, we note that in the reviewer’s comment
>
> > (referred to as "double dipping" in this paper).
>
> the expression **“double dipping”** appears only in our previous submission to another venue and **does not appear anywhere in the current version** of the paper.
>
> We kindly ask the reviewer to reconsider their overall rating, as **all of their concerns have already been addressed in the current version of our submission**:
>
> ---
>
> # 1. Practical necessity of separate modeling
>
> >  the practical necessity of explicitly modeling two separate causal graphs remains somewhat unclear.
>
> In the current version, we have substantially strengthened the practical motivation by providing **three real-world examples** in lines 47–59 in Section 1, namely in systems biology, economics, and algorithmic fairness. These examples illustrate concrete scenarios in which separating mean and variance causal graphs is crucial for interpretability and downstream decision-making.
>
> > Intuitively, one could always first estimate a moment-agnostic causal graph and then ...
>
> In lines 482–485, we now clarify that our approach can achieve **better sample efficiency** and **more principled uncertainty quantification** than approaches that
> rely on previously inferred moment-agnostic causal graphs, including the reviewer's suggested two-stage approach.
>
> # 2. Literature review
>
> > Are there existing models (e.g., beyond HNM) that allow more general functional forms, such as multiplicative or non-additive noise structures ...?
>
> In the current version, we explicitly address this question and state that the answer is **no** by comparing other identifiable SCM classes called PNLs (lines 117–120):
>
> > *Although post-nonlinear models (PNLs) are more general than ANMs and can capture multipricative noise unlike HNMs, they also assume homoscedasticity and thus cannot detect the source of variability.*
>
> Regarding the question on the Gaussian noise assumption,
>
> > Do any existing methods relax the Gaussian noise assumption?
>
> in **Appendix E.6**, where additional experiments on non-Gaussian synthetic datasets are presented, we **added a dedicated discussion** in the description of the baselines  (lines 1393–1397):
>
> > *We compare the performance of our method with ICDH and HOST, both of which are designed for Gaussian HNMs. This is because **most existing methods for non-Gaussian HNMs only address
> bivariate setup (Immer et al., 2023; Tran et al., 2024). An exception is the SkewScore method by Lin et al. (2025), whose official implementation, however, is not publicly available.***
>
> # 3. Identifiablity conditions
>
> > Are the stated conditions only sufficient, or are they also necessary for identifiability?
>
> We have clarified this point explicitly in line 187:
>
> > *By restricting possible structures in this way, we derive the **sufficient (but not necessary)** conditions:*
>
> Regarding the question on unidentifiable scenarios:
>
> > For instance, could the authors give simple concrete examples where identifiability fails (e.g., two models with different graphs but same distribution) ...?
>
> as sketched in lines 194-195, the results by Khemakhem et al. (2021) provide concrete scenarios of non-identifiable cases for HNMs (Theorem B.1 in Appendix B.1.1).
> Non-identifiable cases arise in **linear Gaussian HNMs** or **in HNMs with polynomial variance functions with degree 2 or less**. Therefore, we explicitly exclude these non-identifiable regimes by imposing nonlinearity and piecewise conditions (lines 804–806).
>
>
> # 4. Minor comments
>
> > The title “Moment Matters” may be slightly misleading to readers expecting techniques involving higher-order moments (e.g., skewness, kurtosis), when in fact the method focuses on first and second moments only (mean and variance). May consider rephrasing or clarifying this early in the introduction.
>
> We have clarified this point early in the introduction (lines 28-29):
>
> > *such edges do not specify which statistical moments are affected
> by each cause, thereby limiting interpretability in complex systems that exhibit heteroscedasticity, where different causes may influence the **(conditional) mean and variance** of a variable.*
>
> Regarding the comment on causal minimality:
>
> > Assumption 3.2 (causal minimality) can be explained more in details in the main body, especially to emphasize that it is weaker than the standard faithfulness assumption.
>
> due to space constraints, we could not fully elaborate on this point in the main body. However, in the current version we clarify this in Appendix A (lines 717–723), and we plan to provide a more detailed explanation in the main text if additional space is available in the final version.

---

> ### Comment · Reviewer_LbVq · 2025-11-24
>
> Dear authors and AC,
>
> Thank you for the note.  I do have reviewed for this paper before.  But I did not simply copy paste my prior review.  I have read and compared the current version against the previous one and did not find substantial changes in the main technical contributions (e.g., the whole Sections 3 and 4).  Based on that, I formulated my current review with reuse from previous one.  (I apologize for missing the "double dipping" part in introduction, and appreciate the authors' revision on this)
>
> As a side note, my current score is already reflecting my evaluation of the current version, which is higher than the score I gave in my previous review.  Given that, I thought reusing the comments could also help make authors' response process easier.  If this is deemed inappropriate, please feel free to disregard my evaluation. I apologize.

---

> > ### Author Response · Authors · 2025-11-25
> > **Appreciation and clarifications for Reviewer LbVq**
> >
> > Dear Reviewer LbVq,
> >
> > First, thank you very much for taking the time to review our work—**twice**. We truly appreciate the opportunity your comments gave us to clarify how the resubmission improves upon the prior version. We sincerely apologize if any of our earlier messages came across as complaining; our intention was only to explain the situation carefully.
> >
> > We would like to gently clarify one point:
> >
> > > my current score is already reflecting my evaluation of the current version, which is **higher than the score I gave in my previous review** ...
> >
> > Our understanding is that the current score is the **same** as your (original and final) overall rating in the previous venue (both “marginally above/borderline-accept”). We would like to share this understanding for completeness; we can confirm this point in our view of OpenReview at the previous venue.
> >
> > Since the earlier review, we have strengthened the paper in areas you highlighted as the **strengths of our work in your review**—**uncertainty quantification over causal graphs**:
> >
> > - **Comparison with exact posterior**. Following the feedback by Reviewer YN5K, we now compare our inferred posterior with the exact posterior on synthetic linear-Gaussian datasets. Despite its model complexity, **our method consistently outperforms DDS** (recent variational method) and **achieves competitive performance to MC3** (traditional Bayesian method tailored to linear Gaussian cases), indicating improved fidelity in capturing posterior uncertainty.
> >
> > - **Real-world case study on Sachs dataset**. We added a practical case study on the Sachs dataset to illustrate our moment-driven approach can be used to detect biologically plausible variance-level cause in practice. This analysis was completed **after** you determined (original or final) score at the previous venue.
> >
> > Moreover, we are confident that our current submission has resolved all of your raised "Weaknesses" in your review.
> > If you feel that the current manuscript still has other unresolved issues, we would be grateful to hear them and are happy to provide further clarification or additional experiments. If, on the other hand, you find all the concerns addressed, we would kindly ask whether you might reconsider the overall rating in light of these substantial improvements.
> >
> > Thank you again for your time and thoughtful engagement.
> >
> > Best regards,
> >
> > Authors of Paper #11873

---

> ### Author Response · Authors · 2025-11-27
> **Kind request for overall rating reconsideration**
>
> Dear Reviewer LbVq,
>
> Following the Program Chairs’ guidance, we have privately shared with the Area Chair an anonymized copy of your prior review. Our understanding is that its overall rating,
>
> > 4: **Borderline accept**: Technically solid paper where reasons to accept outweigh reasons to reject, e.g., limited evaluation. Please use sparingly.
>
> maps to your current rating,
>
> > 6: **marginally above the acceptance threshold**. But would not mind if paper is rejected
>
> which appears **unchanged** in effect.
>
> If, upon reading the revised manuscript, you intended to reflect a higher evaluation than in the prior round, we would be very grateful if you could consider **revising the overall rating** in light of the substantive changes in the current version.
>
> For convenience, we briefly summarize how the current submission addresses your points:
>
> - **Strengths, reinforced**: Your noted strength—**Bayesian inference with uncertainty quantification**—is now further supported by new experiments that compare our inferred posterior to the **exact posterior** on linear-Gaussian datasets, as well as a **real-world case study** on the Sachs dataset.
>
> - **Weaknesses, addressed**: The concerns raised in your review have been incorporated and addressed in the current version (details are cross-referenced in our author response).
>
> We hope to avoid further escalations to the AC/PCs if possible, and we truly appreciate your time and consideration. If there is any additional clarification or evidence we can provide, we are happy to do so.
>
> Thank you again for your time and consideration.
>
> Best regards,
>
> Authors of Paper #11873

---

> ### Comment · Reviewer_LbVq · 2025-11-27
>
> Dear authors,
>
> Thank you for the comments.  I apologize to have misremembered my original rating (I thought it was a weak reject, which was indeed somehow my first feeling about this paper).  And I would like to clarify again that my current review and score (6) are accurately reflecting my evaluation of the current version.  This paper is clearly below my bar of a score of 8.  Reasons as follows.
>
> First, the practical necessity of separating mean and variance causal graphs is still doubtful to me (though this does not constitute a deduction in my score).
>   - In particular, the cost of all the strict assumptions (shared causal ordering; nonlinear mean function; piecewise variance function; Gaussian noise) is high. It would be nice though to make these assumptions more explicit in the paper; e.g.,
>     - About the role of Gaussianity, in your response to Reviewer eWXa you mentioned "non-Gaussian noise ... may instead capture any other higher-order moment influences";  then what if non-Gaussian noise is allowed whereas their mean and variance changes are guaranteed? Will the identifiability hold then?
>     - About the nonlinear function, is there anything required about their continuity and k-th order differentiability?
>     - About the piecewise function, does it need to be continuous? Does it need to be positive everywhere as in Khemakhem et al. 2021?
>   - The authors argue that their separation "can achieve better sample efficiency and more principled uncertainty quantification". But note that
>     - For the "sample efficiency", it is due to the introduction of prior knowledge, instead of the separation itself. Also, it doesn't make much sense to talk about "sample efficiency" without providing any finite-sample analysis.  (I'm not asking for such analysis; just about the wording).
>     - For the "uncertainty quantification", it is again due to the Bayesian causal discovery formulation, instead of the separation itself.
>   - In practice, I can imagine that separating the two causal models (e.g., "those are variables that only affect the variance but not the mean") may lead to more misleading artifacts than interpretability. For example,
>     - Thresholding/penalizing the effects on both becomes an issue;
>     - Variance change may also be sensitive to data aggregation, transformations (log, standardizations);
>     - Needing to develop two sets of intervention rules and counterfactual responses, etc.
>
>     If I were a user, I would still prefer to use a single graph representation followed by regression analysis.
>   - Despite all these, I fully acknowledge that my assessment on "practicability" or "strict assumptions" can be subjective. Therefore, **this part does not constitute a deduction in my score.** I still rate for weak acceptance because, under this doubtful and restrictive setting, the authors have successfully conducted comprehensive experiments and derived meaningful results.
>
> The major reason I feel reluctant to raise my score to 8 is that, even ignoring the setting/motivation but focusing on the technical contributions, the two main parts (identifiability and variational inference) do not show much thing new. **They are relatively incremental from existing work in a natural way:**
>   - For the identifiability (of Gaussian HNMs), Theorem 3.5 in this paper is almost the same as Theorem 1 in Yin et al., 2024. The technical progressing is clear: from the abstract condition in Duong & Nguyen 2023 to the explicit condition (in terms of functional forms) in Yin et al., 2024; then, from the so-called "identifiability" in Yin et al., 2024 to the "separability" in this paper, it is fairly simple, if not trivial.
>   - For the variational inference framework, after showing the separability of the two causal graphs, to identify/quantify them, the authors just apply the existing techniques to this setting, such as differentiable DAG sampling in Charpentier et al., 2022, and the overall MLE framework in Yin et al., 2024.  I did not see any new issues identified and addressed in a nontrivial way.
>
> ---
>
> Again, my current score and review already reflect my assessment of the current version.
>
> If the authors are still unsatisfied about not receiving an 8, please feel free to "escalate to the AC/PCs" as suggested.  In doing so, please kindly attach the PDFs of both submissions so the AC/PCs can judge how "substantial", as the authors claim, the changes truly are.

---

> ### Author Response · Authors · 2025-11-28
> **Appreciation, apologies, and responses to the concerns (1/3)**
>
> Dear Reviewer LbVq,
>
> Thank you very much for your detailed and thoughtful response. If any of our earlier messages caused frustration, we sincerely apologize. Our intention was **not** to push for a score of 8, but to clarify whether your overall rating truly reflects your current evaluation, as the suggested rating increase in your message did not appear to be reflected. To proceed appropriately, we confirmed with the Program Chairs that we could privately share your prior review with the Area Chair and then did so; we apologize if this caused any discomfort.
>
> Below we respond to the two main parts of your message. We believe each point is addressable, and we appreciate the opportunity to clarify.
>
> ---
>
> # 1. On assumptions, “sample efficiency,” and possible artifacts
>
> ## 1-1. Assumptions and their clarifications
>
> > the cost of all the strict assumptions (shared causal ordering; nonlinear mean function; piecewise variance function; Gaussian noise) is high
>
> We agree there is an inherent trade-off between **assumption strength** and **identifiability guarantees**,
> as can be seen from comparison between the abstract condition in Duong & Nguyen 2023 and the explicit condition in Yin et al., 2024.
>  In our setting, we chose conditions that make the guarantees explicit and checkable, rather than abstract, so that practitioners can apply the method with clear criteria. Concretely:
>
> - **Shared causal ordering**: This requirement ensures that the **moment-agnostic graph remains acyclic**, which— to the best of our knowledge— is assumed in **all** HNM-based identifiability results.
>
> - **Nonlinear mean** & **piecewise variance functions**: These follow Yin et al. (2024). Assuming nonlinear (mean) functions is common in causal discovery (e.g., nonlinear ANMs). We already note positivity/smoothness (e.g., positivity around line 144; twice-differentiability (and hence continuity) in App. B.1.1) and will highlight them more in the main body in the final version.
>
> - **Gaussian noise**: In our reply to Reviewer eWXa, we discussed the case where **mean/variance functions meet their identifiability conditions** but **noise is non-Gaussian**. In such cases, the adjacency $A^V$ becomes *partially* identified as encoding influences on the **second and potentially higher** moments—i.e., edges indicate **non-mean-level** effects, but attribution to “variance only” may be **ambiguous**.
>
> ## 1-2. What we meant by “sample efficiency” and principled uncertainty
>
> **This point is especially important, so we would like to clarify it further**.
>
> As the reviewer correctly notes, wording "sample efficiency" was somewhat confusing.
> We meant to say that our method **improves data efficiency by avoiding data splitting while maintaining principled uncertainty quantification** (via arguments based on (weak) consistency).
>
> A two-stage approach that first learns a single graph and then separates it into mean/variance can suffer from **data double-dipping**, which is known in statistics to risk **biased estimates** and **over-confident uncertainty** when the same data are reused across tasks.
> This incurred estimation bias is yielded by *overfitting to the data* and by the *overconfidence* in the first-stage inference results (i.e., single graph representation).
>
> - **Difficulty in fair empirical comparison with two-stage approaches**. Note that it is generally difficult to **fairly and meaningfully** demonstrate the extent of this bias empirically, because performance of two-stage procedures depends sensitively on design choices (e.g., regression models, whether the first stage uses point estimates, how data are split).
> - **Theory**: Supporting evidence exists for related work: independence-testing-based methods [1] (two-stage procedure that fits regression models and then conducts an independence test) and the HNM-based one [2; Section 4.2], implying only ideal estimators (e.g., based on Gaussian kernel) can achieve consistency.
>
> In summary,
>
> - **Data recycling** (i.e., two-stage approach) risks biased estimates and over-confident uncertainty, thus harming uncertainty quantification.
>
> - **Data splitting** can mitigate this estimation bias but reduces the effective sample size for each task, thus decreasing data efficiency.
>
> This practical requirement motivates a **single, jointly Bayesian inference** for the separated graphs.
> This is precisely why we argue for an **inference method designed for graph separation**.
> Although we have shortened this point in the current version, we will clarify it in the final version, using additional space.
>
> > [1] Joris M. Mooij, Jonas Peters, Dominik Janzing, Jakob Zscheischler, Bernhard Schölkopf; Distinguishing Cause from Effect Using Observational Data: Methods and Benchmarks. JMLR, 2016.
>
> > [2] Alexander Immer, Christoph Schultheiss, Julia E Vogt, Bernhard Schölkopf, Peter Bühlmann, Alexander Marx. On the Identifiability and Estimation of Causal Location-Scale Noise Models. ICML, PMLR 202:14316-14332, 2023

---

> ### Author Response · Authors · 2025-11-28
> **Appreciation, apologies, and responses to the concerns (2/3; cont'd)**
>
> ## 1-3. On the risk of misleading artifacts
>
> > separating the two causal models (e.g., "those are variables that only affect the variance but not the mean") may lead to more misleading artifacts than interpretability.
>
> Three points help here:
>
> - **Modeling & thresholds**: We model edge probabilities for **each** graph (mean/variance) as separate Bernoulli variables, enabling **graph-specific sparsity control** and principled thresholding.
>
> - **Scope**: Transformations (e.g., strict log transforms) can violate the additive noise assumption underlying HNMs. As noted in our prior response, no identifiable class can address both non-additive and heteroscedastic noise. Such settings are **outside the scope of all HNM-based methods** (including prior work). Our recommendations follow that scope.
>
> - **Needing to develop two sets of intervention rules**: As highlighted in Motivating Example in Introduction, developing such rules is **effective** for supporting variance-sensitive decisions. By preparing intervention rules for mean and variance, we can **effectively search for interventions that achieve desired outcomes with low risk (variance)**. This is a practical advantage of separating the two graphs, not a source of artifacts per se.
>
> ---
>
> # 2. On novelty (identifiability and method) and two-graph challenges
>
> ## 2-1. Novelty of identifiability results
>
> > For the identifiability (of Gaussian HNMs), Theorem 3.5 in this paper is almost the same as Theorem 1 in Yin et al., 2024.
>
> Functionally, we indeed **build on Yin et al.**, with small differences (e.g., non-constancy conditions for variance functions) to ensure **separability**.
>
> The **new** part is the **graphical side**: we prove that **additional shared-order conditions** are needed so that the **moment-agnostic** graph remains **acyclic** and the **mean/variance graphs are separable**. Prior work focuses on identifiability of a **single** (moment-agnostic) graph; the conditions for separating mean and variance had not been formalized.
>
> We do not claim the result is conceptually surprising—only that it is **necessary, explicit, and, importantly, actionable** for our inference procedure.
> That said, our goal is **not** to expand the class of identifiable SCMs per se by introducing surprising conditions,
> but to **provide a theoretically grounded framework for separating mean and variance graphs**.
>
> For this reason, we find that clarifying the practical necessity of graph separation (as in our response to your first part) to the reviewer is much more important. We misunderstood that our prior comments could convince the reviewer on this point, so we sincerely appreciate this opportunity to clarify it further.
>
> ## 2-2. Challenges of two-graph separation
>
> >  the authors just apply the existing techniques to this setting, such as differentiable DAG sampling in Charpentier et al., 2022, and the overall MLE framework in Yin et al., 2024. I did not see any new issues identified and addressed in a nontrivial way.
>
> A central **new issue** with two-graph separation is **scalability**, since the search space grows substantially.
>
> Our contribution is to **align inference with the identifiability structure** by **sharing the permutation across the two graphs** and by **injecting prior knowledge on pairwise orderings** directly into the **differentiable permutation distribution** (Section 4.3). Concretely, leveraging the fact that the permutation is sampled from this distribution by sorting its parameter values, we introduce a **projected-gradient scheme** that enforces inequality constraints on the permutation distribution parameters (Eq. (10)). This **constrained permutation parameterization** effectively shrinks the search space and is **not** in the original differentiable DAG-sampling literature.
>
> Empirically, this idea yields **practical gains**: in the single graph inference on Sachs (n = 100), adding pairwise ordering constraints ("no directed path  $X_j \rightarrow \dots \rightarrow X_i$") improves SHD more than naive edge-absence constraints ("no directed edge $X_j \rightarrow X_i$"), as shown in the table below (included in our response to Reviewer YN5K).
>
> | Method   | No prior knowledge  | 25 \% Edge Absence | 25 \% Pairwise Orderings | 50 \% Edge Absence | 50 \% Pairwise Orderings |
> | -------- | ------------------- | --------------------- | --------------------------- | --------------------- | --------------------------- |
> | Proposed | 16.0 $\pm$ 0.3 | 15.6 $\pm$ 0.3  | **14.9 $\pm$ 0.4**  | 14.1 $\pm$ 0.3   | **13.2 $\pm$ 0.5**         |
> | MC3 | 22.6 $\pm$ 1.1 | 19.8 $\pm$ 0.8  | - | 18.7 $\pm$ 0.7   | - |
> | DDS | 17.6 $\pm$ 0.7 | 17.0 $\pm$ 0.3  | - | 15.9 $\pm$ 0.4   | - |
> | ICDH | 20.0 $\pm$ 1.0 | 19.5 $\pm$ 0.9  | - | 18.5 $\pm$ 0.5   | - |
> | HOST | 16.1 $\pm$ 0.5 | 15.4 $\pm$ 0.2  | - | 14.0 $\pm$ 0.4   | - |

---

> > ### Author Response · Authors · 2025-11-28
> > **Appreciation, apologies, and responses to the concerns (3/3; cont'd)**
> >
> > These results support the claim that order-level priors are strictly more informative than simple edge masks and that our mechanism is effective in practice
> > to  enable more accurate graph recovery **than the baselines that are tailored to single graph inference**, in the presence of ordering-based knowledge.
> >
> > Given the availability of prior knowledge on node orderings in practice (also noted in existing work; line 355), we believe developing such a novel, prior knowledge incorporation in our setup is a significant practical contribution.
> >
> > ---
> >
> > Once again, we apologize if our earlier messages caused frustration. We are grateful for your careful engagement, and we hope the clarifications above are helpful. We are happy to provide any further clarifications, especially on Point 1-2, which we find particularly important for the reviewer's consideration.
> >
> > Best regards,
> >
> > Authors of Paper #11873

---

### Official Review · Reviewer_gKQK · 2025-10-26

**Soundness:** 3
**Presentation:** 3
**Contribution:** 2
**Rating:** 6
**Confidence:** 3

**Summary:**

This paper proposes a Bayesian causal discovery approach that explicitly disentangles and recovers mean and variance causal graphs. Building on heteroscedastic noise models (HNMs), the authors establish identifiability conditions for mean and variance causal graphs and propose a variational inference approach that enables uncertainty quantification. The method is evaluated on synthetic, semi-synthetic, and real-world biological datasets.

**Strengths:**

1. The proposed approach can qualify uncertainty, which is an improvement over most prior point estimation methods.

2. The method allows domain knowledge to be flexibly incorporated into the learning process via a differentiable relaxation of the permutation matrix. This is important for practical small-sample applications.

3. The authors provide comprehensive experimental results to validate the effectiveness of the proposed appoach.

**Weaknesses:**

My main concern is that the novelty of this paper is somehow limited.

- Theorem 1 in (Yin et al., 2024) has already established identifiability of HNM under three conditions. Theorem 3.5 in this paper is very similar to Theorem 1 in (Yin et al., 2024) and the proof of the former relies heavily on the latter.
- While the exploration on mean/variance graph separation is well-motivated, the variational Bayesian treatment, use of Gumbel-Softmax relaxations, and exploitation of domain knowledge via permutation matrix regularization are closely related to existing Bayesian DAG learning frameworks such as DDS (Charpentier et al., 2022), MC3 (Giudici and Castelo, 2003), BayesIMP [1], and BayesDAG [2]. The transition to mean/variance-specific graphs is meaningful, but primarily an extension rather than a conceptual breakthrough.

[1] Bayesimp: Uncertainty quantification for causal data fusion. NeurIPS 2021

[2] Bayesdag: Gradient-based posterior inference for causal discovery. NeurIPS 2023.

**Questions:**

There is a typo in line 179: "if $\pi (i) < \pi (j)$, then $X_{\pi (j)}$ cannot have a directed path to $X_{\pi (i)}$", it seems that $X_{\pi (j)}$ should be "$X_ j$ cannot have a directed path to $X_ i$".

---

> ### Author Response · Authors · 2025-11-21
> **Response to Reviewer gKQK**
>
> We appreciate the reviewer's thoughtful comments from the perspective of novelty, which is essential to improve the paper. Below, we address the reviewer's concerns and suggestions.
>
> ---
>
> # 1. Novelty on the theoretical results
>
> > Theorem 1 in (Yin et al., 2024) has already established identifiability of HNM under three conditions. Theorem 3.5 in this paper is very similar to Theorem 1 in (Yin et al., 2024)
>
> We would like to clarify that identifying the **mean and variance causal graphs** requires two components:
>
> 1. Identifiability of the **moment-agnostic causal graph**, and
> 2. Separability of the **mean and variance causal graphs**.
>
> The novelty of our theoretical results lies in the **second component**, not the first. In particular, we establish the following additional conditions:
>
> - **Graphical conditions** (Assumptions 3.3 and 3.4): The mean and variance graph structures are DAGs with a shared causal order, which ensures that the induced moment-agnostic causal graph is also a DAG.
> - **Functional condition** (Condition (B)): A non-constancy condition on the variance functions, which ensures the identifiability of the variance causal graph structure.
>
> These conditions are **not present** in (Yin et al., 2024), as their work focuses solely on the identifiability of the moment-agnostic causal graph (the first component).
>
> > the proof of the former relies heavily on the latter
>
> The difference in the proof therefore appears in the part dealing with the **separability of the mean and variance causal graphs**, which is presented in **Appendix B.2**. As sketched in lines 196–199, we formally show that the **data joint distributions cannot coincide if either the mean or variance causal graph structures differ**, by leveraging the fact that conditional Gaussian distributions are fully characterized by their means and variances.
>
> Since this separability argument does not appear in (Yin et al., 2024), we would like to emphasize that our theoretical contribution is **novel**: it establishes, from a theoretical standpoint, the possibility of discovering **moment-dependent** causal graph structures (i.e., separate mean and variance causal graphs).
>
>
> # 2. Novelty on the methodological aspects
>
> > the variational Bayesian treatment, use of Gumbel-Softmax relaxations, and exploitation of domain knowledge via permutation matrix regularization are closely related to existing Bayesian DAG learning frameworks such as DDS (Charpentier et al., 2022) ... The transition to mean/variance-specific graphs is meaningful, but primarily an extension rather than a conceptual breakthrough.
>
> We appreciate the reviewer’s recognition of the importance of our work on mean/variance graph separation.
>
> Regarding the methodological aspects, we would like to clarify that our main conceptual contribution lies in **developing an inference method that is theoretically grounded in our identifiability conditions**.
> Concretely, we design the likelihood model and its neural network parameterization so that they **satisfy the graphical and functional conditions** required by our theory. For example, we extend DDS to enforce the shared-order DAG structure on both the mean and variance graphs, thereby aligning the inference procedure with the identifiability conditions.
>
> Unlike in standard Bayesian causal discovery settings, our primary objective is **not** to **introduce novel individual components** (e.g., a new variational family or a new relaxation technique), **nor** to **improve general-purpose Bayesian DAG learning performance**. Instead, our goal is **to enable the discovery of separate mean and variance causal graphs**, which existing methods do not address.
>
> We therefore view the **mean/variance graph separation itself**—together with its theoretically grounded inference procedure—**as the main conceptual breakthrough**. No existing (Bayesian) causal discovery method tackles the problem of identifying distinct mean and variance causal structures, despite its practical importance, which the reviewer also kindly acknowledges.
>
> # Other comments
>
> > There is a typo in line 179: "if $\pi(i) < \pi(j)$, then $X_{\pi(j)}$ cannot have a directed path to $X_{\pi(i)}$", it seems that $X_{\pi(j)}$ should be "$X_{\pi(j)}$ cannot have a directed path to $X_{\pi(i)}$".
>
> We appreciate the reviewer's careful review! We will correct this typo in the revised manuscript. We hope that our responses have addressed all of your concerns.

---

> > ### Comment · Reviewer_gKQK · 2025-11-26
> >
> > Thanks for your clarification. According to my understanding,
> > 1. Your Theorem 3.5 is indeed based on Theorem 1 in Yin et al., 2024 that proves identifiability of the moment-agnostic causal graph. Building upon it, you prove separability of the mean and variance causal graphs.
> > 2. You combined existing techniques to solve your problem: enable the discovery of separate mean and variance causal graphs.
> >
> > Taking everything into consideration, I'd like to maintain my positive score.

---

> > > ### Author Response · Authors · 2025-11-27
> > > **Appreciation**
> > >
> > > We sincerely thank the reviewer for their thoughtful and careful assessment of our work.
> > >
> > > We are glad to hear that both of the points raised are accurate and that our responses have addressed the concerns. We appreciate the reviewer’s engagement and constructive feedback.

---

### Official Review · Reviewer_eWXa · 2025-10-27

**Soundness:** 3
**Presentation:** 2
**Contribution:** 2
**Rating:** 4
**Confidence:** 3

**Summary:**

The paper introduces a mean-variance heteroscedastic noise model (HNM) that separates causal influences on a variable’s mean and variance via two graphs, $G^M$ and $G^V$, assumed to share a topological order. This model is a reparameterization of the original HNM that allows for more interpretable causal discovery. The authors prove identifiability of both graphs under standard HNM assumptions plus the shared ordering constraint, and develop a variational inference method using Gumbel-Softmax and SoftSort to learn both graphs jointly. Experiments on synthetic and real datasets show the method can uncover causal links missed by standard approaches, especially variance-only effects.

**Strengths:**

a. The paper establishes identifiability of dual causal graphs ($G_M$ and $G_V$) under clear conditions. The proof extends existing heteroscedastic causal discovery theory (e.g. Yin et al., 2024) to show that one can recover not just the overall DAG structure but also which edges belong to mean vs. variance relationships, given a shared ordering.

b. It proposes a Bayesian variational approach to infer two linked DAGs simultaneously. The formulation cleverly uses a shared permutation (ordering) for both graphs and generalizes differentiable DAG sampling (DDS) to handle two adjacency matrices. Techniques like Gumbel-Softmax and SoftSort are employed to maintain differentiability, which is an innovative extension of prior continuous DAG optimization methods.

c. The framework naturally incorporates prior knowledge (e.g. known partial ordering of nodes) into the inference procedure. This is valuable for real applications where domain knowledge about causal ordering exists and can guide the search.

**Weaknesses:**

a. The identifiability and method rely on the assumption that the mean and variance causal graphs share a single topological ordering. In practice, this means no cause-effect relationship flips between mean and variance graphs (an edge present in one cannot appear reversed in the other). It might be too restrictive in some real systems. If the true causal mechanism violates this shared order (e.g. a variable influences another’s variance but is downstream in mean effects), the current approach may struggle or require the user to know and enforce the correct ordering upfront.

b. Like standard HNM, the theory requires Gaussian noise and specific nonlinear forms (mean functions must be nonlinear; variance functions non-constant piecewise). These assumptions are crucial for identifiability but limit generality.

c. It seems that most distributions under this two-graph model could also be captured by a standard single-graph HNM, with appropriate functional form (mask). The mean-variance HNM, while more interpretable, does not expand the class of distributions that can be represented compared to the original HNM.

**Questions:**

a. How critical is the Gaussian noise assumption in practice? Could the method be adapted for non-Gaussian heteroscedastic noise in a way that still yields at least partial identifiability?

b. The approach assumes a shared causal order. If this assumption is mildly violated in reality (for instance, one edge slightly conflicts between mean and variance ordering), how robust is the inference?

c. Given the two-graph sampling scheme, what are the practical limits on the number of nodes $d$ the method can handle? Have the authors considered any heuristics or structure in the permutation search (besides simple priors) to improve scalability?

---

> ### Author Response · Authors · 2025-11-21
> **Response to Reviewer eWXa (1/2)**
>
> We sincerely thank the reviewer for the valuable and detailed comments, which help us improve the quality of our work.
>
> ---
>
> # 1. Gaussian noise assumption:
>
> > **Q-a.** How critical is the Gaussian noise assumption in practice? Could the method be adapted for non-Gaussian heteroscedastic noise in a way that still yields at least partial identifiability?
>
> ## 1-1. Theoretical necessity
>
> We impose the Gaussian noise assumption to ensure the **separability** of the moment-agnostic causal graph into the mean and variance causal graphs (lines 473–475).
>
> When the noise is non-Gaussian, the DAG adjacency matrix $A^V$ does not necessarily represents a variance causal graph: an edge $X_i \rightarrow X_j$ may instead capture any other higher-order moment influences from $X_i$ to $X_j$. In this sense, our theory only guarantees **partial** identifiability under non-Gaussian heteroscedastic noise, and does not fully achieve our original goal of identifying moment-dependent causal graph structures.
>
> ## 1-2. Empirical performance
>
> If one is interested solely in the **moment-agnostic** causal graph (rather than explicitly separating mean and variance causal graphs), our method can be adapted to non-Gaussian heteroscedastic noise settings, since identifiability for the non-Gaussian case has been established in prior work (e.g., [1]). As noted in lines 475–477, we have already demonstrated the empirical performance of our method in such cases in Appendix E.6, where **our method outperforms baselines in inferring the moment-agnostic causal graph under non-Gaussian heteroscedastic noise**.
>
> > [1] Eric V. Strobl and Thomas A. Lasko. Identifying patient-specific root causes with the heteroscedastic noise model. Journal of Computational Science, 72:102099, 2023.
>
> # 2. Shared causal order condition
>
> > **Q-b.** The approach assumes a shared causal order. If this assumption is mildly violated in reality (for instance, one edge slightly conflicts between mean and variance ordering), how robust is the inference?
>
> As described in lines 180–183, the combination of the shared causal order and the DAG conditions on both mean and variance causal graphs ensures that **the induced moment-agnostic causal graph is itself a DAG**, which is essential for identifiability.
> To see this, consider the case where an edge $X_i \rightarrow X_j$ appears in the mean graph but the opposite edge $X_i \leftarrow X_j$ appears in the variance graph (i.e., the orderings conflict). The resulting moment-agnostic causal graph necessarily contains a cycle, and thus identifiability fails.
>
> To the best of our knowledge, **no existing work establishes identifiability on the HNMs with such cyclic causal graph structures**. As the reviewer points out, complex real systems may indeed exhibit cyclic structures in their moment-agnostic causal graphs. However, handling these cases goes beyond the current theoretical framework of HNM-based methods and remains an open problem. We therefore regard this as an important but out-of-scope direction for future research, rather than a limitation specific to our approach.
>
> # 3. Practical heuristics for improving scalability
>
> > **Q-c.** Given the two-graph sampling scheme, what are the practical limits on the number of nodes $d$ the method can handle? Have the authors considered any heuristics or structure in the permutation search (besides simple priors) to improve scalability?
>
> In a high-dimensional setup with $d=50$ (a typical regime in existing work), our method outperforms the baselines for both **moment-dependent** and **moment-agnostic** causal graph inference (Figures 2, 4, and 5).
>
> To further improve scalability in permutation search,
> one promising direction is to exploit **temperature annealing heuristics** for tuning the hyperparameter, called temperature $\tau$ (in Eq. (30)), used to control the smoothness of the differentiable permutation distribution. This hyperparameter controls the smoothness of the differentiable permutation distribution:
>
> - Larger $\tau$ yields a smoother optimization landscape but a looser approximation to discrete permutations for computing the gradients.
> - Smaller $\tau$ yields a sharper, more discrete-like approximation but can make optimization more difficult.
>
> Temperature annealing is a widely used heuristic, where training starts with a smoother distribution (large $\tau$) and gradually decreases $\tau$ to obtain nearly discrete permutations toward the end of training. Due to its complication of tuning procedure, we do **not** employ such annealing in our experiments and instead fix $\tau=1.0$ (the PyTorch default) for reproducibility. However, we observed that varying $\tau$ can change the average SHD by around 1–2 on mean-variance HNM datasets with $d=10$, depending on the dataset. This suggests that temperature annealing could serve as a practical strategy for effective optimization, thus further improving scalability in high-dimensional settings.

---

> > ### Author Response · Authors · 2025-11-21
> > **Response to Reviewer eWXa (2/2; cont'd)**
> >
> > # Other comments
> >
> > > **W-c.** It seems that most distributions under this two-graph model could also be captured by a standard single-graph HNM, with appropriate functional form (mask). The mean-variance HNM, while more interpretable, does not expand the class of distributions that can be represented compared to the original HNM.
> >
> > We thank the reviewer for their careful review!
> >
> > Exactly as the reviewer states, the class of distributions represented by both HNMs are **identical**. However, we would like to emphasize that our main goal is not to expand the expressivity of identifiable HNMs, **but to improve their interpretability** for elucidating complex real-world systems exhibiting heteroscedasticity.
> >
> > We hope that our responses can revolve all the reviewer's concerns.

---

> > > ### Author Response · Authors · 2025-11-25
> > > **Further clarifications to Reviewer eWXa**
> > >
> > > We apologize for our repeated responses, but we would like to provide further clarifications regarding the reviewer's central question:
> > >
> > > > If this assumption is mildly violated in reality (for instance, one edge slightly conflicts between mean and variance ordering), how robust is the inference?
> > >
> > > **Answer.** In the absence of identifiability guarantees for **cyclic** causal graph structures, **we do not believe there is a technically sound way to quantify the robustness** of graph inference under such violations.
> > >
> > > Naively, we could consider some synthetic data experiments where the ground-truth mean and variance causal graphs slightly violate the shared-order DAG condition and demonstrate how the (expected) SHD score decreases when the number of edge conflicts increases.
> > > However, **such experiments would be confusing and potentially misleading to the community**,
> > > because the **identifiability under cyclic structures is generally non-trivial** and **highly depends on the SCM functional forms and the noise distributions** (e.g., the functional relationships under the *equilibrium state* of cyclic SCMs [1]).
> > >
> > > For this reason, we prefer **not** to include experiments that violate the shared-order DAG assumption by creating mean–variance conflicts, as they would implicitly rely on **model-specific, non-established identifiability** in cyclic regimes.
> > > We view the question of identifiability for cyclic HNMs as **an important open problem that lies beyond the scope of the present work**. Once principled identifiability results for cyclic HNMs are established, we would be eager to investigate how our moment-driven approach can be extended and to revisit robustness in that setting.
> > >
> > > We hope this clarification addresses the reviewer's concern, and we remain happy to discuss this point further—along with any other aspects of the paper.
> > >
> > > > [1] Mooij, Joris M and Janzing, Dominik and Heskes, Tom and Schölkopf. On causal discovery with cyclic additive noise models. In NeurIPS, 2011.

---

> > > > ### Comment · Reviewer_eWXa · 2025-11-26
> > > >
> > > > Thank you for the authors' reply. Most of my questions have been answered. However, I still have some doubts about the contribution of this paper. Based on the answer and my understanding, while the mean-variance HNM is easier to interpret, it does not expand the range of representable distributions compared to the original HNM. Instead, this paper merely attempts to represent it in a different way, which does not substantially improve its identifiability.

---

> ### Author Response · Authors · 2025-11-26
> **Response to Additional Comments from Reviewer eWXa**
>
> We sincerely appreciate the reviewer taking the time to respond to our comments and for asking further questions that help clarify key theoretical aspects of our work.
>
> # On the separability of moment-agnostic graph into moment-dependent graphs
>
> > Instead, this paper merely attempts to represent it in a different way, which does not substantially improve its identifiability.
>
> We would like to emphasize that representing the **same class of distributions** using **moment-dependent causal graph structures** (separate mean and variance graphs in this work) is **not a trivial reparameterization** from a theoretical standpoint. In general, **establishing separability**—i.e., proving that the data joint distributions **cannot coincide** when any of moment-dependent causal graph structures differ—**is highly non-trivial**.
>
> As we noted in our response to Reviewer gKQK, we prove that **additional functional and graphical conditions** (including shared-order condition) **are required for such separability**, leveraging the fact that conditional Gaussian distributions are fully characterized by their means and variances (see **Appendix B.2** for the proof).
>
> In short, **expressivity is preserved**, but deriving these additional conditions for the decomposition **clarifies and sharpens what can be identified** under the stated assumptions.
>
> # Beyond Gaussianity: a future direction
>
> An interesting direction—also raised by the reviewer—is to establish **separability conditions** under **non-Gaussian** heteroscedastic noise. As discussed in lines 487-489, the main challenge is to **define an SCM subclass** in which *higher-order moment graphs* (e.g., for skewness and kurtosis) can be represented in a way that **guarantees separability**. With the current mean–variance HNM, separability can fail for the following reason:
>
> - **Ambiguity under non-Gaussian heteroscedastic noise**. As responded in our previous message, in non-Gaussian settings the “variance graph” may **inadvertently capture higher-order moment influences**, **not purely variance**. Disentangling variance from higher moments would require extending the model and its assumptions.
>
> Although such an extension would **not** expand the set of representable joint distributions per se, it would still be **practically valuable**, as it could illuminate **why** certain variables exhibit skewness or heavy tails in real systems—thereby enhancing interpretability and decision support.
>
> > Based on the answer and my understanding, while the mean-variance HNM is easier to interpret, it does not expand the range of representable distributions compared to the original HNM.
>
> We understand the reviewer’s concern: our goal differs from standard work that primarily seeks to enlarge the identifiable class of distributions. However, we are confident that **this shift toward moment-level identifiability and interpretability** is **a significant step** for causal discovery in complex real-world systems.

---

> > ### Author Response · Authors · 2025-11-27
> > **Additional clarifications on contributions**
> >
> > > However, I still have some doubts about the contribution of this paper.
> >
> > Since we are unsure whether our earlier responses fully addressed the reviewer’s concerns about our contributions, we would like to offer the following clarifications.
> >
> > # Motivation: Why separating mean and variance matters in practice
> >
> > Our work is **not** motivated by a simple modification of HNMs.
> > As detailed in lines 47–59 of the Introduction, we identify **three real-world settings** where separating mean and variance causal graphs is crucial for interpretability and downstream decisions: systems biology, economics, and algorithmic fairness.
> >
> > We find the variance-level analysis particularly compelling for **algorithmic fairness**.
> > As noted in econometrics (e.g., [1]), humans often care about **risk** in outcomes (e.g., hiring, lending) and may (inadvertently) make decisions based on attributes that influence the variance of outcomes. If those attributes are themselves causally affected by (possibly unobserved) sensitive features (e.g., race and gender), the resulting decisions can produce **unfair treatment** of certain demographic groups. Crucially, such **variance-level statistical discrimination often goes unnoticed** [1], making it a **latent** form of unfairness. Our moment-driven causal discovery framework aims to reveal these latent variance pathways, supporting **forward-looking fairness** in building reliable predictive models.
> >
> > However, standard causal discovery methods—by inferring a **single, moment-agnostic graph**—cannot identify variance-level causes. This practical gap is the motivation for our work.
> >
> > > [1] David L. Dickinson and Ronald L. Oaxaca. Statistical discrimination in labor markets: An experimental analysis. Southern Economic Journal, 76(1):16–31, 2009.
> >
> > # Theoretical contribution: Novelty of our identifiability results
> >
> > Our contribution is to provide a **theoretically grounded framework** that supports the above applications.
> > As clarified in earlier responses, our theoretical novelty lies in proving the **separability of mean and variance causal graphs** under **additional graphical and functional conditions**. These conditions are **not present** in prior work (e.g., Yin et al., 2024), which targets identifiability of **a moment-agnostic graph only**.
> >
> > In short, we preserve expressivity while introducing conditions that **sharpen what can be identified** at the moment level.
> >
> > # Methodological contribution: Inference aligned with identifiability
> >
> > We develop a Bayesian inference procedure that is **aligned with these identifiability conditions**.
> > As the reviewer kindly noted in the “strengths,” our method cleverly shares the permutation (topological order) between the mean and variance graphs, thus **ensuring the shared-order DAG requirement from our theory is satisfied at inference time**.
> >
> > This alignment between theory and inference is a key **strength**: in causal discovery, it is not uncommon for inference procedures to be designed **without** careful attention to the identifiability conditions they implicitly assume.
> >
> > Empirically, as the reviewer also observes, our method uncovers causal links missed by standard approaches—including a biologically plausible **variance-only** cause in the Sachs dataset.
> >
> > ---
> >
> > **Summary**: Our work is driven by a **clear practical need**, and we propose a **theoretically grounded framework** for separating mean and variance causal graphs. We view this as a **significant step** toward addressing variance-sensitive decisions in complex real-world systems.

---

### Official Review · Reviewer_YN5K · 2025-10-31

**Soundness:** 3
**Presentation:** 3
**Contribution:** 3
**Rating:** 4
**Confidence:** 3

**Summary:**

The paper introduces a Bayesian causal discovery method for mean-variance hierarchical noise model causal graphs: where the mean and variance of a node depends on its parents. Crucially, the mean and variance graphs can be different.

The authors provide an identifiability result and provide a variational method for recovering an approximate posterior using neural networks to approximate the mean and variance functions. Some improvements in the implementation are introduced to make the optimization more tractable, and the method is compared against several competing Bayesian methods on a variety of data sets.

**Strengths:**

+ The identifiability result is, as far as I can tell, correct and nicely motivated.
+ The parameterization is well-structured and clearly chosen in a way that makes the variational inference tractable.
+ The implementation tricks such as the two-phase optimization and prior knowledge incorporation are novel applications to improve the performance of the approach.
+ The empirical results are quite strong on these small datasets.

**Weaknesses:**

+ I am not sure how effectively the proposed probabilistic model can actually capture the posterior of the mean-variance HNM. As I understand it, the mean and variance functions are represented by an MLP which is independent of $A^M$ and $A^V$ (except insofar as the edges are masked out with the adjacency matrices when applying the MLP). Therefore, the MLP would have to learn mean and variance functions that would be applicable when the underlying inputs correspond to different nodes. As a somewhat trivial example, if we had some posterior density on $X_1 \rightarrow X_2$ and also on $X_1 \leftarrow X_2$, then the MLP would have to be able to represent the mean function from $X_1$ to $X_2$, but also from $X_2$ to $X_1$. It's possible I have misunderstood this point, but it seems like this independence assumption could be quite limiting.

+ The contribution of the prior knowledge incorporation is a bit unclear, since there's no baseline for the other methods. It's hard to tell if the improvement is just because of the fact that some of the edges are being specified to correctly exist/not exist.

**Questions:**

+ Is it possible to adjust your method to condition the MLP on the permutation or adjacency matrix?
+ Could you provide some small-dimensional (2,3,4)-node cases where the posterior can be calculated exactly or via direct sampling, and compare this approach to the exact posterior?
+ Could you provide a comparison to the other methods when they are given the prior knowledge incorporation? You could for instance take the methods' solution and specify that an edge must exist in the solution when given the prior information.
+ Did you experiment with different values of the temperature parameters $\tau$? How does the solution accuracy/optimization smoothness trade-off?

typos: 'VARINACE' in table 4.

---

> ### Author Response · Authors · 2025-11-21
> **Response to Reviewer YN5K (1/2)**
>
> We thank the reviewer for the suggestions regarding the technical soundness of our posterior inference and prior-knowledge incorporation, which are very helpful for improving the quality of our work. We are also pleased that the reviewer finds the strengths of our theoretical claims and empirical results.
>
> ---
>
> # 1. Modeling of MLPs and Causal DAG Adjacency Matrices
>
> > As I understand it, the mean and variance functions are represented by **an MLP** which is independent of $A^M$ and $A^V$.
>
> We would like to clarify this point. We use **different MLPs for different nodes**, and we always evaluate the likelihood of the parameters of $2d$ MLPs $\Theta = \{(\theta^M_j, \theta^V_j)\}_{j=1}^{d}$ (for $d$ nodes) conditional on the **sampled DAG adjacency matrices** $A^M$ and $A^V$. Mean and variance functions $m_j$ and $v_j$ for the $j$-th node are given by the $j$-th pair of MLPs, $m_j(A^M_j \odot X; \theta^M_j)$ and $v_j(A^V_j \odot X; \theta^V_j)$, whose inputs are masked according to the $j$-th column vectors of the sampled adjacency matrices $A^M$ and $A^V$.
>
> Therefore, in the reviewer's example:
>
> > if we had some posterior density on $X_1 \rightarrow X_2$ and also on $X_1 \leftarrow X_2$,
>
> mean adjacency matrix $A^M$ drawn from the estimated posterior can be $X_1 \rightarrow X_2$ **or** $X_1 \leftarrow X_2$, depending on the sampled graph.
> We evaluate how well each node-specific MLP represents the mean function $m_j(\cdot)$ for node $X_j$ ($j=1, 2$) **conditional on the sampled $A^M$**; hence,
>
> - When sampled $A^M$ has edge $X_1 \rightarrow X_2$,
>   MLP $m_2$ is trained to express the mean function from $X_1$ to $X_2$.
> - When sampled $A^M$ has edge $X_1 \leftarrow X_2$,
>   MLP $m_1$ is trained to express the mean function from $X_2$ to $X_1$.
>
> Therefore, we never force the **same MLP** to simultaneously express both functional relationships underlying $X_1 \rightarrow X_2$ and $X_1 \leftarrow X_2$; instead, different node-specific MLPs are responsible for these directions,
> depending on the sampled DAG adjacency matrices.
>
> > Is it possible to adjust your method to condition the MLP on the permutation or adjacency matrix?
>
> **Yes**. We always condition the likelihood evaluation of the MLP parameters $\Theta$ on the sampled permutation and the sampled adjacency matrices.
> To evaluate the objective function in Eq. (8),
> we approximate its expectation over the causal DAGs using Monte Carlo sampling
> by computing the log likelihood of $\Theta$ in the expectation with masked inputs according to the sampled causal DAGs.
>
> We will clarify in the revised manuscript that we train $2d$ MLPs (for the mean and variance functions over $d$ nodes) and explicitly explain how the inputs are masked using the mean and variance causal DAG adjacency matrices.
>
> # 2. Posterior approximation performance evaluation
>
> > Could you provide some small-dimensional (2,3,4)-node cases where the posterior can be calculated exactly or via direct sampling, and compare this approach to the exact posterior?
>
> We appreciate the reviewer’s constructive suggestion. Since exact posterior computation is only feasible for tractable model classes, we consider **linear Gaussian ANM synthetic datasets** (with sample size $n=250$). We compare the performance of inferring posteriors over **moment-agnostic** causal graphs
> with the two Bayesian methods, MC3 and DDS.
>
> As the distance between the inferred and exact posteriors, we use the **total variation distance** (values in [0, 1]):
>
> | Method   | d = 2  | d = 3 | d = 4 |
> | -------- | ------------------- | --------------------- | --------------------------- |
> | Proposed | 0.26 $\pm$ 0.06 | 0.21 $\pm$ 0.09  | **0.34 $\pm$ 0.08**        |
> | MC3 | **0.01 $\pm$ 0.00** | **0.10 $\pm$ 0.01**  | 0.42 $\pm$ 0.10        |
> | DDS | 0.31 $\pm$ 0.10 | 0.32 $\pm$ 0.08  | 0.45 $\pm$ 0.11        |
>
> and the **KL divergence** (non-negative):
>
> | Method   | d = 2  | d = 3 | d = 4 |
> | -------- | ------------------- | --------------------- | --------------------------- |
> | Proposed | 1.01 $\pm$ 0.46 | 0.34 $\pm$ 0.09  | **1.51 $\pm$ 0.28**        |
> | MC3 | **0.01 $\pm$ 0.00** | **0.09 $\pm$ 0.01**  | 6.52 $\pm$ 0.91        |
> | DDS | 1.94 $\pm$ 0.28 | 1.81 $\pm$ 0.21  | 5.52 $\pm$ 0.73       |
>
> (Both metrics are averaged over 10 random seeds; smaller is better.)
>
> **Our method consistently outperforms DDS** (recent variational method), despite the model complexity increase from homoscedastic to heteroscedastic Gaussian likelihoods.
> Since MC3 (traditional Bayesian method) is specifically designed for linear Gaussian ANMs, it achieves the best posterior approximation performance when $d = 2, 3$.
> However, it becomes worse when $d=4$ due to its poor scalability.
> Moreover, as shown in Figures 2, 3, 4, and 5, this method performs poorly on nonlinear datasets.
>
> We will add these small-dimensional evaluation results to the revised version.

---

> > ### Author Response · Authors · 2025-11-21
> > **Response to Reviewer YN5K (2/2; cont'd)**
> >
> > # 3. Performance comparison on prior knowledge incorporation
> >
> > > The contribution of the prior knowledge incorporation is a bit unclear, since there's no baseline for the other methods. It's hard to tell if the improvement is just because of the fact that some of the edges are being specified to correctly exist/not exist.
> >
> > We appreciate this helpful comment. A key point is that a pairwise node ordering constraint $\pi(i) < \pi(j)$ ($\pi$: permutation over node orders) is strictly more informative than simply forbidding the directed edge $X_j \rightarrow X_i$:
> > the ordering constraint rules out **all directed paths** $X_j \rightarrow \dots \rightarrow X_i$, including but not limited to the direct edge $X_j \rightarrow X_i$.
> >
> > > Could you provide a comparison to the other methods when they are given the prior knowledge incorporation? You could for instance take the methods' solution and specify that an edge must exist in the solution when given the prior information.
> >
> > Following the reviewer’s valuable suggestion, for each method, we evaluate the performance of **a naive variant** that only uses edge-absence constraints: it additionally masks out the edge  $X_j \rightarrow X_i$ when a pairwise node constraint $\pi(i) < \pi(j)$ is given, but does **not** constrain the permutation itself. The SHD metric score on the Sachs dataset (with sample size $n=100$) are:
> >
> > | Method   | No prior knowledge  | 25 \% Edge Absence | 25 \% Pairwise Orderings | 50 \% Edge Absence | 50 \% Pairwise Orderings |
> > | -------- | ------------------- | --------------------- | --------------------------- | --------------------- | --------------------------- |
> > | Proposed | 16.0 $\pm$ 0.3 | 15.6 $\pm$ 0.3  | **14.9 $\pm$ 0.4**  | 14.1 $\pm$ 0.3   | **13.2 $\pm$ 0.5**         |
> > | MC3 | 22.6 $\pm$ 1.1 | 19.8 $\pm$ 0.8  | - | 18.7 $\pm$ 0.7   | - |
> > | DDS | 17.6 $\pm$ 0.7 | 17.0 $\pm$ 0.3  | - | 15.9 $\pm$ 0.4   | - |
> > | ICDH | 20.0 $\pm$ 1.0 | 19.5 $\pm$ 0.9  | - | 18.5 $\pm$ 0.5   | - |
> > | HOST | 16.1 $\pm$ 0.5 | 15.4 $\pm$ 0.2  | - | 14.0 $\pm$ 0.4   | - |
> >
> > As expected, **encoding prior knowledge at the level of node orderings** consistently yields better performance than merely enforcing edge absence, supporting the usefulness of our proposed mechanism beyond simply “specifying some edges to correctly not exist”.
> >
> > # 4. Effects of temperature parameter
> >
> > > Did you experiment with different values of the temperature parameters $\tau$?
> >
> > We thank the reviewer for this suggestion. Throughout our experiments, we fix the temperature parameters to $\tau = 1.0$ (the PyTorch default) to ensure reproducibility. However, we observed that it slightly affects the causal graph inference performance: for instance, on synthetic, mean-variance HNM datasets with $d = 10$ nodes, increasing its value to $\tau=3.0$ reduces the average SHD by about 1–2, though its optimal value depends on the dataset.
> >
> > > How does the solution accuracy/optimization smoothness trade-off?
> >
> > As is standard for Gumbel-Softmax–type relaxations, there is a trade-off:
> >
> > - Larger $\tau$ leads to a smoother optimization landscape but a coarser (less accurate) approximation of the discrete sampling.
> > - Smaller $\tau$ yields a sharper, more accurate approximation to discrete permutations/adjacency matrices, at the cost of a less smooth optimization landscape and potentially higher variance in gradients.
> >
> > For this reason, a **temperature annealing heuristic** is often employed in differentiable distribution learning: $\tau$ is gradually decreased during training to shift from smooth approximate permutations to nearly discrete ones. Such annealing could improve stability and the final performance, but it complicates hyperparameter tuning (initial $\tau$, final $\tau$, and schedule), which we aimed to avoid in our experiments for clarity and reproducibility.
> >
> > We will clarify in the camera-ready version that our main reported results use a fixed
> > $\tau=1.0$, and we will briefly summarize the above observations about the trade-off.
> >
> > # Other comments
> >
> > > typos: 'VARINACE' in table 4.
> >
> > We thank the reviewer for their careful reading! We hope that our responses have addressed all of your concerns.

---

> > > ### Author Response · Authors · 2025-11-24
> > > **Point-by-Point Summary for Reviewer: Modeling, Posterior Accuracy, Permutation-based Priors, and Temperature**
> > >
> > > We sincerely thank the reviewer for raising many important questions about the technical soundness of our posterior inference and prior-knowledge incorporation—both are central to our work. Because our answers are necessarily long and may be hard to follow, we summarize the key points below:
> > >
> > > - **What our MLPs represent**: **We use different MLPs for different nodes**, and so each node-specific MLP is trained to represent the mean or variance function for each node, by constructing the input mask vector from the sampled DAG adjacency. Therefore, **we never force a single MLP to simultaneously express both functional relationships underlying $X_1 \rightarrow X_2$ and $X_1 \leftarrow X_2$**; instead, different node-specific MLPs handle these directions.
> > >
> > > - **Posterior approximation performance**: Following the reviewer's insightful suggestion, we conducted the performance comparison on linear Gaussian ANM datasets, where the marginal likelihood (and thus the posterior) can be computed exactly. Compared with the Bayesian baselines (MC3 and DDS), **our method consistently outperforms DDS and achieves better performance than MC3 when $d=4$**, despite its model complexity. These results demonstrate the effectiveness of our method in capturing the inference uncertainty over causal graphs.
> > >
> > > - **Prior knowledge incorporation**: As requested, we compared our permutation-based prior knowledge incorporation with a naive edge-absence masking approach. The results show that **encoding prior knowledge at the level of node orderings consistently yields better performance than merely enforcing edge absence**, supporting the usefulness of our proposed mechanism beyond simply “specifying some edges to correctly not exist”.
> > >
> > > - **Temperature parameter effects**: For reproducibility, we fix the temperature parameter $\tau=1.0$ (the PyTorch default) in all experiments. As with standard Gumbel-Softmax–type relaxations, **there is a trade-off between optimization smoothness and solution accuracy**; in practice, **annealing $\tau$ can improve stability and final performance. We deliberately avoided this additional heuristic for clarity**.
> > >
> > > We are grateful for the reviewer’s thoughtful engagement and hope this summary helps address their concerns. We are happy to provide any further clarification.

---

> > > > ### Comment · Reviewer_YN5K · 2025-11-24
> > > > **Updating review**
> > > >
> > > > Thanks for your response.
> > > > Your clarification on the MLP parameterization was helpful--I can see that I did misunderstand how the permutation matrices are used. The MLP is conditioned on graph due to masking out the non-parents of the node when applying it.
> > > >
> > > > Just to check my understanding, is it the case that for the two fully connected networks with topological order $(X_1, X_2, X_3)$ and topological order $(X_2, X_1, X_3)$, the outputs for node-specific MLP for $m_3$ is the same, since the parents of $X_3$ are $X_1$ and $X_2$ in both cases?
> > > >
> > > > Sorry to focus on a seemingly-minor point, but I think it is quite important to clearly distinguish (as far as possible) the probabilistic assumptions that are imposed when choosing the data generating process (HNM), and the variational family (your parameterization). Thanks to your response, I understand that the variational family is not as factorized as I thought.
> > > >
> > > > I'm glad to see the other additional experimental results. I'm confident in raising my recommendation

---

> > > > > ### Author Response · Authors · 2025-11-25
> > > > > **Response to Additional Questions from Reviewer YN5K**
> > > > >
> > > > > We sincerely appreciate the reviewer taking the time to respond to our comments and for asking further questions that help clarify key technical aspects of our method.
> > > > >
> > > > > > The MLP is conditioned on graph due to masking out the non-parents of the node when applying it.
> > > > >
> > > > > **Answer**: **Exactly**. Given sampled DAG adjacency matrices ($A^M$ and $A^V$), we construct an input mask for each node that zeros out non-parents when feeding inputs to its node-specific MLP (for the mean or variance).
> > > > >
> > > > > > is it the case that for the two fully connected networks with topological order ($X_1, X_2, X_3$) and topological order ($X_2, X_1, X_3$), the outputs for node-specific MLP for $m_3$ is the same, since the parents of $X_3$ are $X_1$ and $X_2$ in both cases?
> > > > >
> > > > > **Answer**: **If the parents of $X_3$ are identical, then yes**.
> > > > >
> > > > > That said,  **the parents of $X_3$ need not coincide across those two orders**, because the adjacency matrices (in Eq. (5))
> > > > >
> > > > > > $A^M = \Pi^{\top} U^M \Pi, \quad A^V = \Pi^{\top} U^V \Pi$,
> > > > >
> > > > > are determined by the sampled permutation matrix $\Pi$ **and** the sampled **upper-triangular binary matrices** $U^M, U^V \in \{0, 1\}^{d \times d}$.
> > > > > Here, $U^M$ and $U^V$ act as **masks that select parental nodes only from earlier nodes in the sampled order.**
> > > > > Thus, even if the same two nodes precede $X_3$ in both sampled orders, **the actual parent set of of $X_3$** (and hence the inputs to its MLP) can differ if the corresponding entries in $U^M$ or $U^V$ differ.
> > > > >
> > > > > >  I think it is quite important to clearly distinguish (as far as possible) the probabilistic assumptions that are imposed when choosing the data generating process (HNM), and the variational family (your parameterization).
> > > > >
> > > > > We completely agree with the reviewer. Regarding the **variational family**, as noted in lines 236-237, we consider a mean-field approximation:
> > > > >
> > > > > > $P(A^M, A^V) = \sum_{U^M, U^V, \Pi} P(U^M) P(U^V) P(\Pi)$,
> > > > >
> > > > > and in practice we adopt a sampling-based scheme that **independently samples** $U^M, U^V$ and $\Pi$ according to this factorization, then forms the DAG adjacency matrices via Eq. (5) in the above.
> > > > > This is **separate from** the modeling assumptions of the HNM data-generating process (which specify functional forms and noise), and we will make this distinction more explicit in the revision.
> > > > >
> > > > > We are grateful that the reviewer increased the overall rating after our clarifications! We hope the answers above address all remaining concerns, and we are happy to provide any additional details or experiments if helpful.

---

### Meta-Review · Area_Chair_M8kp · 2026-01-06

**Summary:**

This paper proposes a Bayesian causal discovery method to disentangle and recover separate mean and variance causal graphs from observational data under a heteroscedastic noise model.

**Reviewer Concerns:**

Although the paper has some merits, such as a theoretically sound identifiability result and a well-structured variational inference framework with practical implementation tricks, the issues raised by the reviews are critical. For instance, the limited and incremental contribution of the paper compared to the original HNM (eWXa), with the identifiability result heavily relying on prior work and the technical approach being a direct extension of existing Bayesian DAG learning frameworks (gKQK), and potential limitations in capturing the true posterior (YN5K). Although the authors address some issues in their responses, the paper still needs to address the remaining concerns from the reviewers.

**Reviewer Scores:**

Reviewer YN5K would raise the score based on the previous discussion.

Reviewer gKQK, LbVq would maintain the score as the neutral stance of the reviews.

Reviewer eWXa would remain the score toward rejection as the concerns remain unsolved.

---

### Decision · Program_Chairs · 2026-01-26

Reject